# Natural variation in *BnaA9.NF-YA7* contributes to drought tolerance in *Brassica napus* L

Jia Wang [1,2], Lin Mao[1,2], Yangyang Li[1,2], Kun Lu [1,2], Cunmin Qu [1,2], Zhanglin Tang [1,2], Jiana Li [1,2] & Liezhao Liu [1,2] ✉

Rapeseed *(Brassica napus)* is one of the important oil crops worldwide. Its production is often threatened by drought stress. Here, we identify a transcription factor (BnaA9.NF-YA7) that negatively regulates drought tolerance through genome-wide association study in *B. napus*. The presence of two SNPs within a CCAAT cis element leads to downregulation of *BnaA9.NF-YA7* expression. In addition, the M63I (G-to-C) substitution in the transactivation domain can activate low level expression of *BnaA4.DOR*, which is an inhibitory factor of ABA-induced stomatal closure. Furthermore, we determine that *Bna.ABF3/4s* directly regulate the expression of *BnaA9.NF-YA7*, and *BnaA9.NF-YA7* indirectly suppresses the expression of *Bna.ABF3/4s* by regulation of *Bna.ASHH4s*. Our findings uncover that *BnaA9.NF-YA7* serves as a supplementary role for ABA signal balance under drought stress conditions, and provide a potential molecular target to breed drought-tolerant *B. napus* cultivars.

*Brassica napus* L. (*B. napus*; AACC, $2n = 38$), an amphidiploid species formed by the natural hybridization of *Brassica rapa* (AA, $2n = 20$) and *Brassica oleracea* (CC, $2n = 18$)[1], is an excellent source of edible vegetable oil and protein and constitutes the second-largest vegetable oil source in the world[2–4]. However, drought stress has a major limiting impact on its production and yield stability[5], and drought accounts for at least a 30% yield loss in *B. napus* every year[6]. Therefore, exploring the mechanisms by which *B. napus* responds to drought stresses and how the resistance of *B. napus* to drought stress may be improved has great significance. To date, some drought-responsive genes, including *LEA3* and *VOC* (which alter drought tolerance and lipid accumulation[2]), *BnaYUC6a* (which is associated with high auxin production and drought tolerance[7]), *BnaA6.RGA* (which enhances drought tolerance[8]), and *BnaA01.CIPK6* (which confers drought tolerance[9]), have been cloned and functionally characterized in *B. napus*. However, the molecular regulation of drought tolerance in *B. napus* is largely unknown.

Nuclear factor-Y (NF-Y), also known as heme activator protein (HAP) or CCAAT-binding factor (CBF), is a sequence-specific transcription factor with nonspecific DNA-binding nucleosome properties that is ubiquitous in eukaryotes, and is composed of NF-YA, NF-YB, and NF-YC subunits[10,11]. The three subunits can form a heterotrimeric complex and bind to the downstream gene CCAAT-box, one of the sequence elements most frequently found in eukaryotic promoters, and regulate its expression[11]. In recent years, many studies have found that NF-Y transcription factors in plants play an important role in plants' ability to cope under various abiotic stresses, including drought stress[12–15]. From a functional viewpoint, NF-Y is considered a facilitator of TF binding and an architectural promoter organizer[11]. Furthermore, NF-Y complex binding to DNA is also important for the establishment of histone posttranslational modifications (PTMs) through the recruitment of relevant enzymes[16].

Here, we report that an NF-YA subunit, BnaA9.NF-YA7, is involved in *B. napus* drought tolerance at the seedling stage. Two SNPs in the promoter and a nonsynonymous substitution (M63I) in *BnaA9.NF-YA7* confer *B. napus* drought tolerance by regulating the ABA signal transduction pathway via feedback inhibition of the expression of *BnaABF3/4s*-related genes.

[1]College of Agronomy and Biotechnology, Southwest University, Beibei, Chongqing 400715, China. [2]Academy of Agricultural Science, Southwest University, Beibei, Chongqing 400715, China. ✉e-mail: liezhao@swu.edu.cn

## Results

### Natural variation in *BnaA9.NF-YA7* is associated with drought tolerance

Further mining of previously published data[17] showed that *BnaA9.NF-YA7* is associated with the germination percentage (GP) and germination index (GI) under drought stress and that *BnaA9.NF-YA7* is upregulated by drought stress under repeated drought-rewatering cycles (Supplementary Fig. 1). To investigate functional allelic variations in the *BnaA9.NF-YA7* locus, 48 variation sites covering the *BnaA9.NF-YA7* promoter (2 kb) and gene were identified based on previously published resequencing data[18] (Supplementary Data 1). Candidate gene association analysis was performed for GP and GI using 181 inbred lines from the resequencing population and five SNPs were identified above the suggestive significance threshold of association ($P < 0.01$; Supplementary Fig. 2a, b). Moreover, GWAS performed in this natural-variation population for water holding capacity (WHC) under drought stress using a mixed linear model approach identified ten associated

loci meeting the threshold for suggestive evidence of association ($P < 1.7 \times 10^{-7}$), of which the most significant signal (S9_19159584) was located within the third exon of *BnaA9.NF-YA7* (Fig. 1a and Supplementary Fig. 2c).

Interestingly, S9_19159584 (G-to-C), in which the methionine at the 63rd position was changed to isoleucine (M63I) in *BnaA9.NF-YA7*, was in strong linkage disequilibrium (LD) with the SNPs within the promoter region (Fig. 1b). The 608 rapeseed genotypes were classified into five haplotype groups (Supplementary Fig. 2d), based on the SNPs within *BnaA9.NF-YA7* (MAF ≥0.03), and physiologic index analysis showed significant differences among different haplotypes, especially Hap3 and Hap4 (Fig. 1c, d and Supplementary Fig. 2e–h). Moreover, the accessions with the C allele (Hap1 and Hap4, referred to as *BnaA9.NF-YA7^C*) had a higher WHC than the accessions with the G allele (referred to as *BnaA9.NF-YA7^G*) after drought treatment (Fig. 1e and Supplementary Fig. 2i). However, no mutation sites significantly associated with drought tolerance were found in another homolog *BnaC5.NF-YA7*

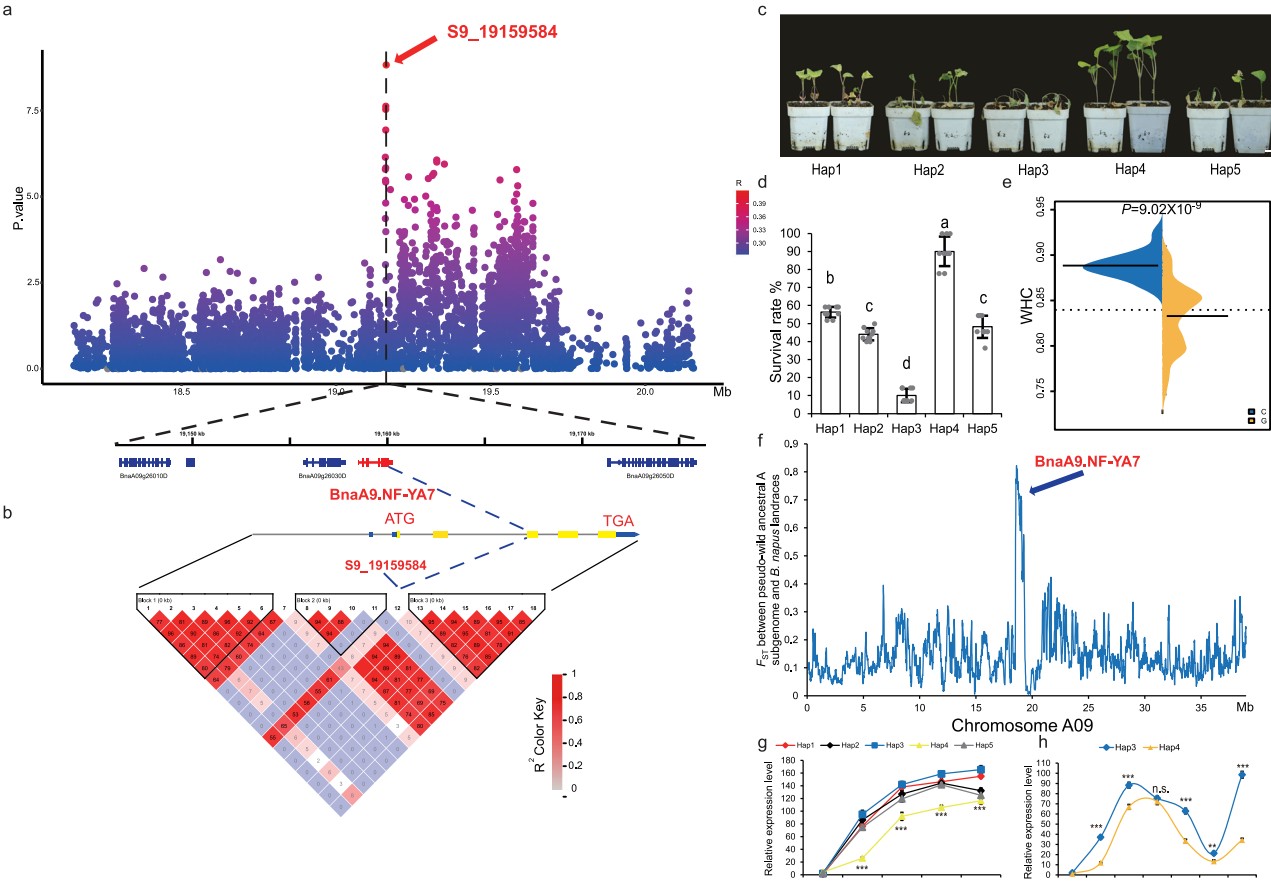

**Fig. 1 | Natural variation in *BnaA9.NF-YA7* is associated with drought tolerance.** **a** Regional Manhattan plot of the *BnaA9.NF-YA7* genomic region on chromosome A09. The 2-Mb genomic region on either side of the most significant SNP is shown. The lead SNP is shown with a red dot, and the other SNPs are colored according to their LD ($r^2$ value) with the lead SNP. **b** Pairwise LD estimates for *BnaA9.NF-YA7* gene locus. The green arrows denote the significantly associated SNP in *BnaA9.NF-YA7*. **c** The different varieties from five haplotypes survived after 14 days of drought treatment followed by 5 days of rewatering. Scale bars = 5 cm. **d** Comparison of survival rates among accessions of the five haplotypes. Values are means ± SD (Hap1: $n = 27$ accessions examined over ten independent experiments; Hap2: $n = 188$ accessions examined over ten independent experiments; Hap3: $n = 14$ accessions examined over ten independent experiments; Hap4: $n = 9$ accessions examined over ten independent experiments; Hap5: $n = 11$ accessions examined over ten independent experiments;). The lowercase letters indicate significant differences ($P < 0.05$, one-way ANOVA followed by two-tailed Fisher's least

significant difference (LSD) test. **e** Bean plot showing the water holding capacity of accessions from two alleles. $P$ values were calculated with the two-tailed Student's $t$-test (C: $n = 21$ accessions; G: $n = 184$ accessions). **f** The level of population differentiation ($F_{ST}$) across chromosome A09 between *B. napus* and *B. rapa*. **g** Expression levels of *BnaA9.NF-YA7* in five haplotypes under 0.2 M D-mannitol treatment. The values are means ± SD ($n = 3$ biological replicates). Asterisks indicate statistically significant differences between Hap3 and Hap4 interaction based on two-tailed Student's $t$-test (***$P < 0.001$ and $P = 4.18 \times 10^{-5}$, $3.50 \times 10^{-4}$, $8.85 \times 10^{-5}$, and $4.20 \times 10^{-4}$ at the time points of 3, 6, 12, and 24 h). **h** Expression levels of *BnaA9.NF-YA7* in Hap3 and Hap4 under 15% PEG treatment. The values are means ± SD ($n = 3$ biological replicates). Asterisks indicate statistically significant differences between Hap3 and Hap4 interaction based on two-tailed Student's $t$-test (**$P < 0.01$, ***$P < 0.001$, and $P = 1.78 \times 10^{-5}$, $5.92 \times 10^{-4}$, $8.71 \times 10^{-5}$, $3.70 \times 10^{-3}$, and $5.78 \times 10^{-6}$ at the time points of 3, 6, 24, 48, and 72 h). n.s. not significant. Source data are provided as a Source Data file.

in *B. napus* (Supplementary Fig. 2c). Together, these results suggest the genetic variation in *BnaA9.NF-YA7* could be involved in the drought tolerance of *B. napus*.

Next, we performed multiple sequence alignments and phylogenetic tree analysis of the BnaA9.NF-YA7 protein sequence. Surprisingly, the genotypes of *Bra.NF-YA7*, *Bol.NF-YA7* and *BnaC5.NF-YA7* at position 63 were consistent with *BnaA9.NF-YA7^C* (Supplementary Fig. 3a), and *BnaA9.NF-YA7^C* was evolutionarily closer to *Bra.NF-YA7* (Supplementary Fig. 3b), indicating that only 10% (52/608) of the C alleles in *B. napus* were from ancestral species. Nevertheless, the M63I substitution did not change the tertiary structure of the protein (Supplementary Fig. 3c). Several *BnaA9.NF-YA7* fragments were designed based on the domain structure of *BnaA9.NF-YA7* (Supplementary Fig. 3d). The Y2H (self-activation) assay and dual-luciferase reporter assay showed that *BnaA9.NF-YA7* had different transcriptional activation abilities in Hap3 and Hap4 (Supplementary Fig. 3e–h), and the M63I substitution occurred in a Pro-rich transcriptional activation domain (Supplementary Fig. 3a). Unlike the Hap1 varieties, the Hap4 members were clustered in a small subgroup of the semi-winter ecotype (Supplementary Fig. 4a). The highest $F_{ST}$ levels appeared at the site in *BnaA9.NF-YA7* and its flank but not in *BnaC5.NF-YA7* (Fig. 1f and Supplementary Fig. 4b), suggesting that *BnaA9.NF-YA7* possibly underwent strong selection during the domestication stage.

### *BnaA9.NF-YA7* was greatly induced by drought and abscisic acid

Under nonstress conditions, higher levels of *BnaA9.NF-YA7* transcripts were found in embryos, leaves, and seeds compared with other tissues, similar to the result for *BnaC5.NF-YA7* (Supplementary Fig. 5a). *BnaA9.NF-YA7* expression was significantly induced by drought stress, and the expression level of Hap3 was significantly higher than that of Hap4 (Supplementary Fig. 5b). The same phenomenon occurred under osmotic stress simulated by 0.2 M D-mannitol and 15% PEG, with a negative correlation between survival rate and expression level, which was almost the same as the result under drought treatment (Fig. 1g, h and Supplementary Fig. 5c, d). Unexpectedly, the expression level of Hap1 was also higher than that of Hap4 (Fig. 1g). Moreover, the expression of *BnaC5.NF-YA7* was also induced by drought, but the level of *BnaC5.NF-YA7* was not significantly different among the five haplotypes (Supplementary Fig. 5e).

Compared to the unaltered transcripts in water-treated leaves (Supplementary Fig. 6a), the expression of *BnaA9.NF-YA7* was upregulated by $H_2O_2$, ABA, and SNP (sodium nitroprusside, nitric oxide producer) treatment, and there were also differences between Hap3 and Hap4 (Supplementary Fig. 6b–d). Likewise, the expression of *BnaA9.NF-YA7* was significantly induced after dehydration treatment (Supplementary Fig. 6e). Upregulation of *BnaA9.NF-YA7* expression by ABA treatment and induction of *BnaA9.NF-YA7* expression was suppressed by pretreatment with FLU (fluridone, an inhibitor of ABA biosynthesis) in detached leaves, suggesting that ABA was an essential signal for dehydration-induced transcription of *BnaA9.NF-YA7* (Supplementary Fig. 6f). These findings indicated that *BnaA9.NF-YA7* is responsive to drought stress and ABA signaling.

### *BnaA9.NF-YA7* negatively regulates *B. napus* drought tolerance

To confirm the function of *BnaA9.NF-YA7*, we generated transgenic *B. napus* plants (Supplementary Fig. 7). The results of drought tolerance evaluation showed that most of the knockout and RNAi seedlings survived after a 5-day recovery because of increased leaf water holding capacity and decreased water loss rate and transpiration rate (Fig. 2a–e and Supplementary Fig. 8a–e). A similar result was also observed for germination under D-mannitol treatment; the knockout seedlings exhibited increased root length and hypocotyl length, but the BnaA9.NF-YA7-OE lines had shorter roots and hypocotyls than the WT (Supplementary Fig. 8f–h). These results demonstrate that *BnaA9.NF-YA7* functions as a negative regulator of *B. napus* drought tolerance.

Furthermore, we noted that there was no significant difference in drought tolerance between the *BnaA9.NF-YA7^C* and *BnaA9.NF-YA7^G*-overexpressing transgenic lines (Fig. 2f), but the drought tolerance of Hap3 could be slightly enhanced by the site-directed mutation M63I (Supplementary Fig. 8i, j).

In comparison with wild-type plants, the aaCC (*BnaA9.NF-YA7* knockout) and aacc (*BnaA9.NF-YA7/BnaC5.NF-YA7* double knockout) lines had smaller stomata and lower stomatal density (Fig. 2g–k), exhibiting lower stomatal conductance and transpiration rate under nonstress conditions (Fig. 2l, m). Conversely, overexpressing *BnaA9.NF-YA7* led to increased stomatal conductance and transpiration rate compared with those of the WT, but significant differences were not found among these transgenic plants after ABA treatment (Fig. 2l, m). Additional confirmation was obtained by using infrared thermal imaging to visualize the surface temperature of detached leaves, in which the leaf surface temperature of the overexpression lines was lower due to more water loss than that of the WT and knockout lines before dehydration treatment (Supplementary Fig. 8a and Supplementary Fig. 9a, b), while that of the knockout line was slightly higher than that of the WT after 2 h of dehydration (Supplementary Fig. 9c). ABA treatment significantly inhibited root and hypocotyl growth in all genotypes (Fig. 2n), and the inhibition degree significantly increased in BnaA9.NF-YA7-OE plants compared to the WT and BnaA9.NF-YA7-KO lines (Fig. 2o, p), indicating that overexpression of *BnaA9.NF-YA7* increased ABA sensitivity. However, the percentage of closed stomata was significantly lower in BnaA9.NF-YA7-OE (19.68%) compared to WT (52.50%) or BnaA9.NF-YA7-KO (62.87%) leaves after drought treatment (Supplementary Fig. 9d, e), similar to the result for ABA treatment (Fig. 2q). These results indicated that *BnaA9.NF-YA7* negatively regulates drought tolerance by increasing stomatal conductance and the transpiration rate.

### CCAAT-box variation affects the expression level of *BnaA9.NF-YA7*

GUS activity was clearly detected in all tissues and organs except for the root tip with different degrees of expression; in particular, GUS staining was more vasculature-specific in seedlings (Supplementary Fig. 10). In 2-week-old plants, GUS activity and GUS expression levels in leaves were significantly different between Hap3 and Hap4 under abiotic stress (Fig. 3a–c). To accurately identify the specific mutation sites that led to the differences in Hap3 and Hap4 expression levels, we conducted promoter truncation analysis and found that the ABRE, CCAAT-box, and motif_sequence were the core cis-acting elements of the *BnaA9.NF-YA7* promoter (Fig. 3d), but the presence of two SNPs led to the CCAAT-box deletion in only Hap4, not in the other haplotypes (Fig. 3e and Supplementary Fig. 11). These findings provide strong evidence that CCAAT-box variation is directly involved in the difference in *BnaA9.NF-YA7* transcript levels between Hap4 and other haplotypes (including Hap1).

To confirm this hypothesis, expression analysis and clustering of all NF-Y subunits (41 putative BnaNF-YA, 43 putative BnaNF-YB, and 18 putative BnaNF-YC subunits) were performed under repeated drought-rewatering treatment, and the NF-YA/YB/YC subunits within the B2 subgroup to which *BnaA9.NF-YA7* belongs were selected for interaction analysis (Supplementary Fig. 12a). Y2H and BiFC assays detected interactions between BnaA9.NF-YB2 and BnaA6.NF-YC9, BnaA9.NF-YB2, and BnaC5.NF-YC9 (Supplementary Fig. 12b, c), and the BnaA9.NF-YB2/BnaA6.NF-YC9 dimer formed a trimer with BnaA6.NF-YA5 and BnaC5.NF-YA9 by yeast three-hybrid (Y3H) analysis (Supplementary Fig. 12d). Next, transient dual-luciferase assays were performed in *Arabidopsis* protoplasts. When co-transforming with an empty effector construct, the LUC activity of the Hap3 reporter was not different from that of the Hap4 reporter (Fig. 3f, g). Co-expression of BnaA9.NF-YB2/BnaA6.NF-YC9 with BnaA6.NF-YA5 induced LUC activity in both the Hap3 and Hap4 reporters, similar to BnaA9.NF-YB2/

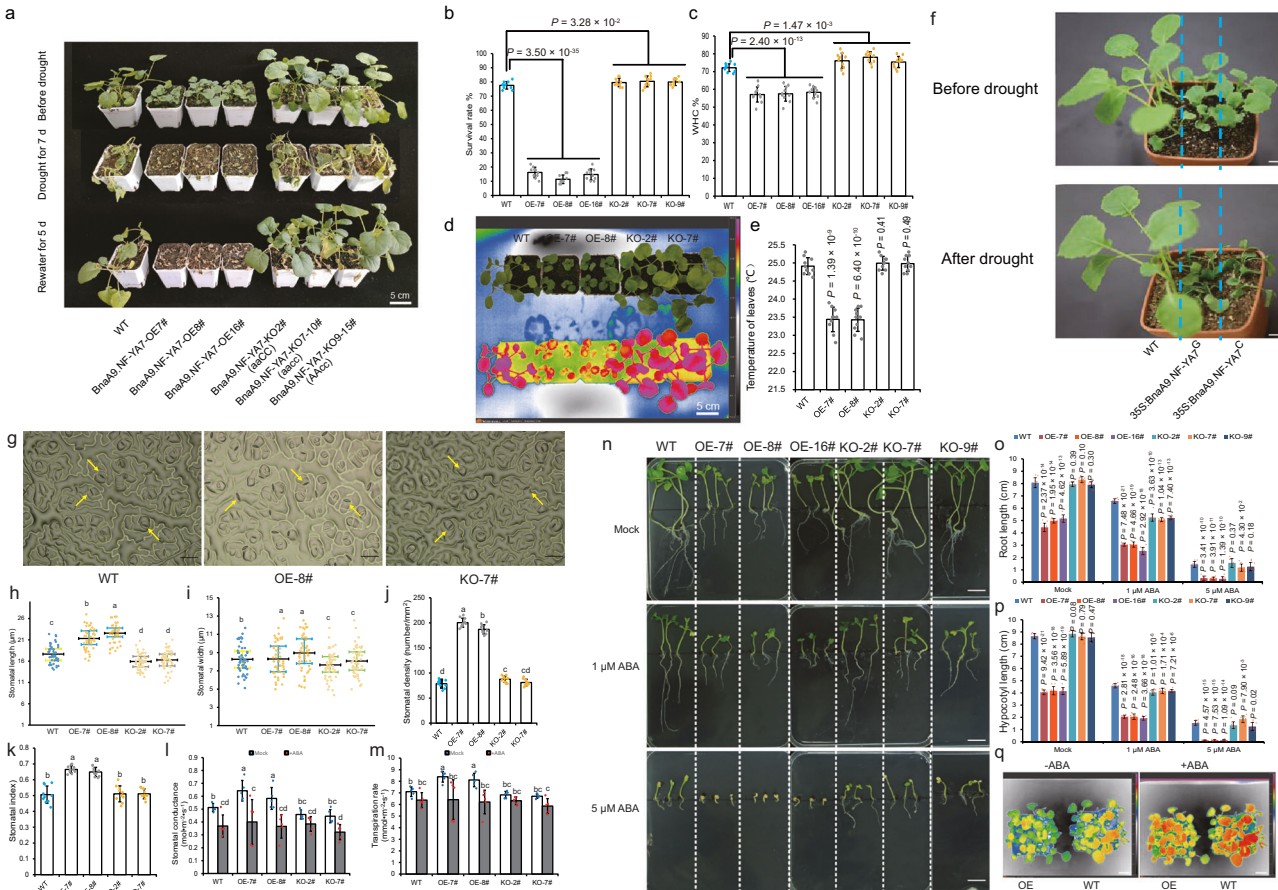

**Fig. 2 | *BnaA9.NF-YA7* negatively regulates *B. napus* drought tolerance by increasing stomatal conductance and the transpiration rate. a** Phenotype of *B. napus* seedlings grown in plastic pots under well-watered, drought stress, or rewatering conditions. Bar = 5 cm. **b, c** The survival rate (**b**) and water holding capacity (**c**) of overexpression and knockout lines were compared with those of WT. The values are means ± SD (*n* = 50 plants examined over ten independent experiments in (**b**) and *n* = 10 biologically independent experiments in (**c**). *P* values were calculated with the two-tailed Student's *t*-test. **d, e** Infrared thermal images show the wild-type and *BnaA9.NF-YA7* transgenic lines (**d**) and the corresponding leaf surface temperature (**e**) before drought treatment. The values are means ± SD (*n* = 10 biologically independent experiments). *P* values were calculated with the two-tailed Student's *t*-test. Bar = 5 cm. **f** The drought tolerance was not significantly different between the 35S:BnaA9.NF-YA7^C and 35S:BnaA9.NF-YA7^G transgenic lines. Bar = 1 cm. **g** Phenotype of stomata between wild-type and *BnaA9.NF-YA7*

transgenic lines under well-watered conditions. Bar = 10 μm. **h–k** Comparison of stomatal length (**h**), stomatal width (**i**), stomatal density (**j**), and stomatal index (**k**) between wild-type and *BnaA9.NF-YA7* transgenic lines. The values are means ± SD (*n* = 50 biological replicates in (**h**, **i**); *n* = 10 biological replicates in (**j**, **k**). **l, m** Stomatal conductance (**l**) and transpiration rate (**m**) before and after ABA treatment. Values are mean ± SD (*n* = 5 biologically independent experiments). The lowercase letters indicate significant differences (*P* < 0.05, One-way ANOVA followed by two-tailed LSD test). **n** Phenotype of different genotypes grown on half-strength MS phytoagar medium containing 0, 1, or 5 μM ABA for 14 days under LD conditions. White bars = 2 cm. **o, p** Quantification of root length (**o**) and hypocotyl length (**p**) after 14 days of ABA treatment. The values are means ± SDs (*n* = 10 biological replicates). *P* values were calculated with the two-tailed Student's *t*-test. **q** Infrared thermal images of WT and BnaA9.NF-YA7-OE plants before and after ABA treatment. White bars = 1 cm. Source data are provided as a Source Data file.

BnaC5.NF-YC9 with BnaC5.NF-YA9, whereas the Hap4 reporter exhibited lower LUC activity than the Hap3 reporter (*P* < 0.001, Fig. 3f, g), which explained the difference in expression levels between Hap3 and Hap4. In addition, we found that BnaA9.NF-YA7 can induce higher LUC activity in the Hap3 reporter than in Hap4, and BnaA9.NF-YA7^G induced higher LUC activity than BnaA9.NF-YA7^C, which further confirms that the M63I substitution is in the transcriptional activation region.

### *BnaABF3/4s* directly regulate the activity of *BnaA9.NF-YA7*

To explore the upstream regulators that regulate *BnaA9.NF-YA7* expression under drought stress, we screened for proteins that interact with D5 (a fragment of the promoter of *BnaA9.NF-YA7*, Fig. 3d) using a Y1H assay with a cDNA library prepared from drought-treated *B. napus* leaves (Supplementary Table 1). Among the potential regulatory genes, *BnaABF3/4s* were particularly interesting because of the upregulated expression of *BnaA9.NF-YA7* induced by ABA. Both the Y1H assay and electrophoretic mobility shift assay (EMSA) showed that

BnaC7.ABF3 and BnaA3.ABF4 bound the ABRE sites in the *BnaA9.NF-YA7* promoter in vitro (Fig. 4a–d). Furthermore, ChIP–qPCR identified enrichment in the fragment spanning the ABRE sites in the *BnaA9.NF-YA7* promoter (Fig. 4e–g). We thus believed it was possible that *BnaC7.ABF3* and *BnaA3.ABF4* regulated the expression of *BnaA9.NF-YA7* under ABA induction.

A previous study suggested that *AtABF3* and *AtABF4* interact in *Arabidopsis*[19]. We tested for an interaction between the BnaC7.ABF3 and BnaA3.ABF4 proteins before further determining how this circuit was regulated, and both the Y2H assay and BiFC assay showed that there was a physical interaction between BnaC7.ABF3 and BnaA3.ABF4 (Supplementary Fig. 13a, b). Transient expression assays showed that overexpressing BnaC7.ABF3 or BnaA3.ABF4 alone induced the LUC activity of both the Hap3 and Hap4 reporters, and co-expression of BnaC7.ABF3 with BnaA3.ABF4 further induced the activity of the two reporters, whereas there was no significant difference in LUC activity between the Hap3 and Hap4 reporters (Fig. 4h, i).

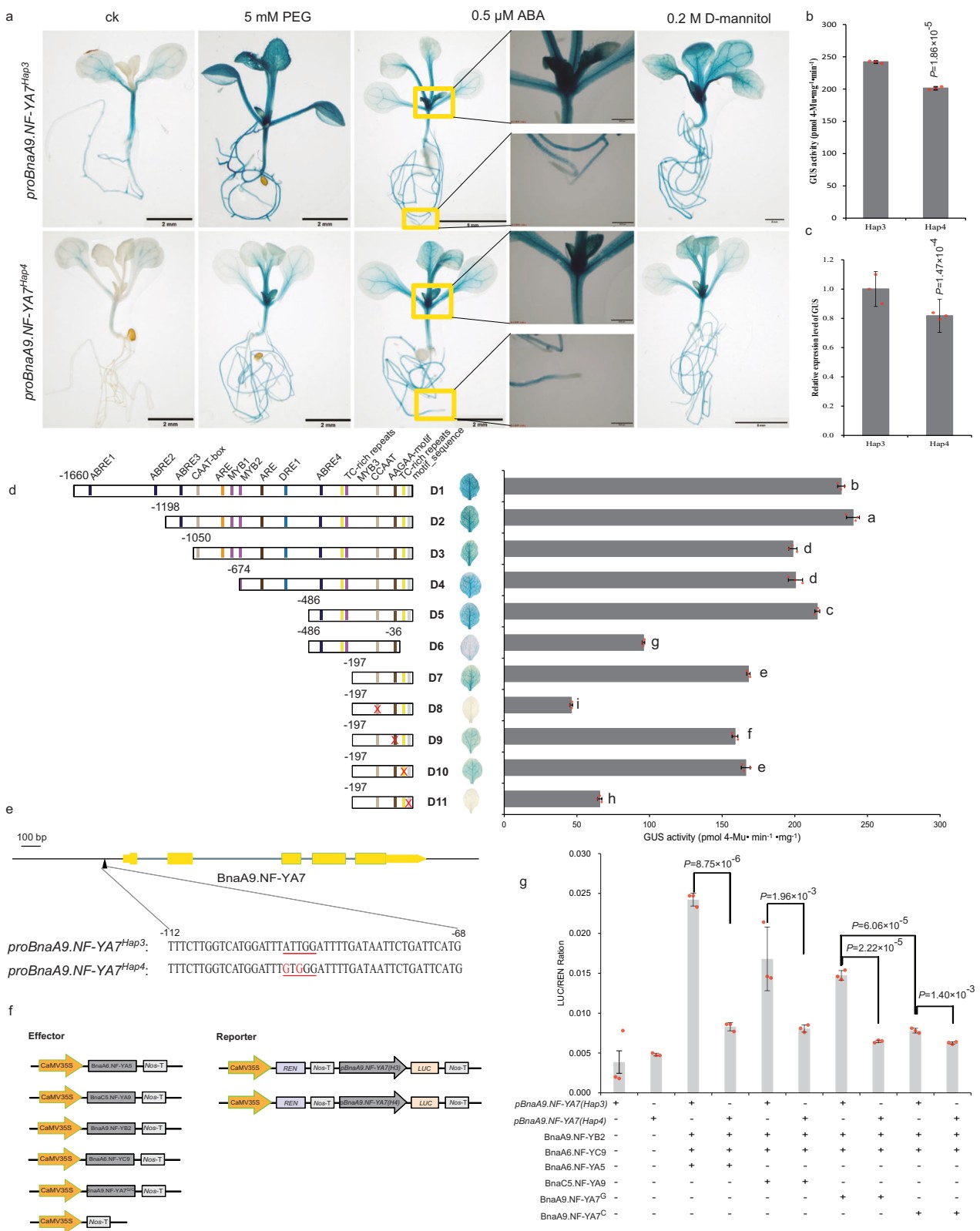

To further determine the regulatory relationship of *BnaA9.NF-YA7* with *BnaC7.ABF3*, we examined their expression in the leaves of *BnaC7.ABF3* transgenic plants. The expression level of *BnaA9.NF-YA7* was significantly increased in the BnaC7.ABF3-OE lines, but the transcript was repressed in the BnaC7.ABF3-RNAi lines even after ABA treatment, whereas the expression of *BnaC7.ABF3* was unaltered in the BnaA9.NF-YA7-RNAi line (Supplementary Fig. 13c). Together,

these data clearly indicate that *BnaA9.NF-YA7* functions downstream of *BnaABF3/4s*.

## Identification of stress-related genes controlled by *BnaA9.NF-YA7*

To understand the different molecular mechanisms regulated by *BnaA9.NF-YA7* under drought stress, we compared the transcriptomes

**Fig. 3 | CCAAT-box variation affects the expression level of *BnaA9.NF-YA7*.**
**a** GUS staining of *ProBnaA9.NF-YA7^Hap3^*:GUS and *ProBnaA9.NF-YA7^Hap4^*:GUS seedlings under normal conditions and 5 mM PEG, 0.5 μM ABA, and 0.2 M D-mannitol treatment. Scale bars, 2 mm, 5 mm or 500 μm. **b, c** GUS enzyme activity (**b**) and GUS expression level (**c**) between *ProBnaA9.NF-YA7^Hap3^*:GUS and *ProBnaA9.NF-YA7^Hap4^*:GUS seedlings under normal conditions. The values are means ± SD (*n* = 3 biological replicates). *P* values were calculated with the two-tailed Student's *t*-test. **d** GUS enzyme activity of transgenic plants after fusion of the GUS reporter gene with *BnaA9.NF-YA7* promoter fragments of different lengths. The values are means ± SDs (*n* = 3 biologically independent experiments). D1–D11 indicates

*BnaA9.NF-YA7* promoter fragments of different lengths. The lowercase letters indicate significant differences (*P* < 0.05, one-way ANOVA followed by a two-tailed LSD test). **e** SNP variation of the CCAAT-box within the promoter region of *BnaA9.NF-YA7* between Hap4 and the other Haps. The sequences on the red lines are the target site. The red characters indicate substitutions. **f** Schematic diagram showing the constructs used in the transient transactivation analysis. **g** NF-Y trimers can activate the expression of *BnaA9.NF-YA7*, and the CCAAT-box variation affects the expression level of *BnaA9.NF-YA7* between Hap3 and Hap4. The values are means ± SD (*n* = 3 biologically independent experiments). *P* values were calculated with the two-tailed Student's *t*-test. Source data are provided as a Source Data file.

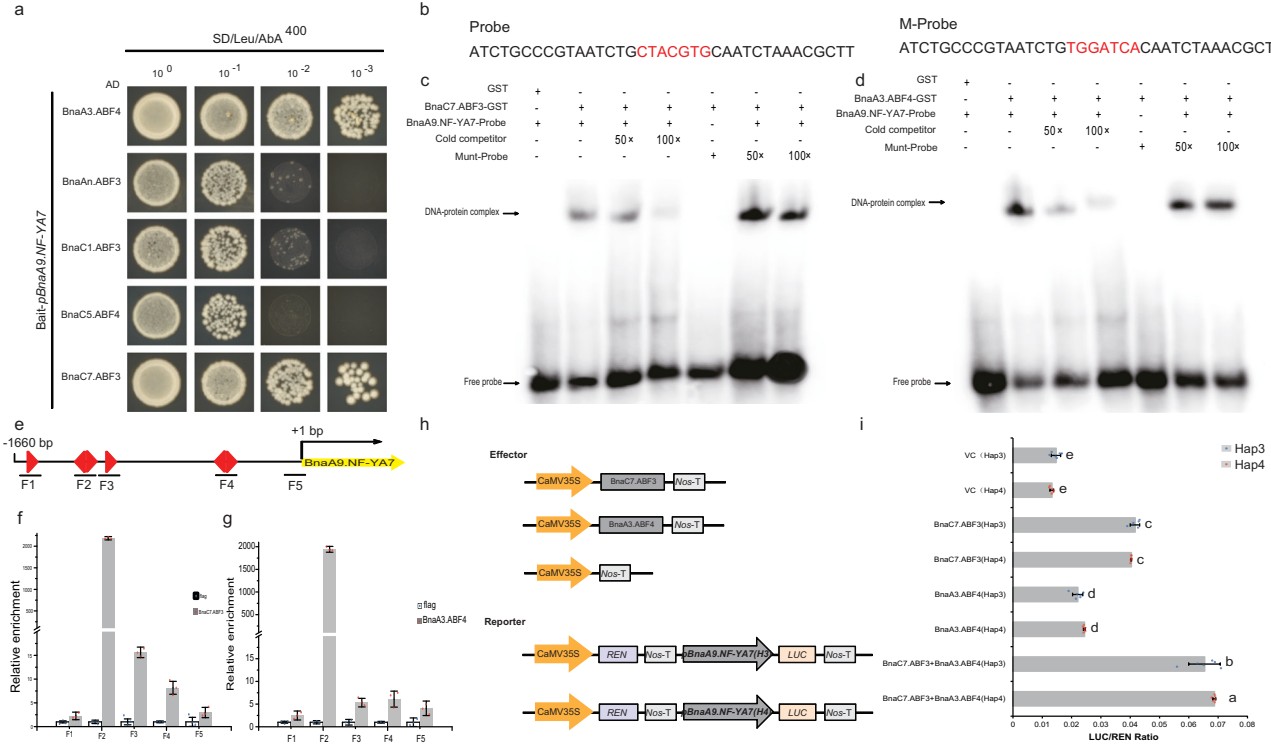

**Fig. 4 | *BnaABF3/4 s* directly upregulates the transcription of *BnaA9.NF-YA7*.**
**a** Y1H indicated that BnaABF3/4 s could bind the *BnaA9.NF-YA7* promoter.
**b**–**d** EMSA showed that BnaC7.ABF3 (**c**) and BnaA3.ABF4 (**d**) could bind the ABRE sites in the *BnaA9.NF-YA7* promoter. The biotin-labeled probes are indicated in (**b**). **e** The fragments used in ChIP–qPCR. **f, g** ChIP–qPCR identified BnaC7.ABF3 (**f**) and BnaA3.ABF4 (**g**) as significantly enriched in the fragment containing the ABRE sites in the *BnaA9.NF-YA7* promoter. Fold enrichments in (**f**) and (**g**) were calculated

relative to input. The values are means ± SD (*n* = 3 biological replicates).
**h** Schematic diagram showing the constructs used in the transient transactivation analysis. **i** BnaC7.ABF3 and BnaA3.ABF4 activates *BnaA9.NF-YA7* independently or in the form of dimers. The values are means ± SDs (*n* = 6 biologically independent experiments). The lowercase letters indicate significant differences (*P* < 0.05, One-way ANOVA followed by two-tailed LSD test). Source data are provided as a Source Data file.

of leaves from drought-stressed 4-week-old Hap3 and Hap4 plants using RNA-Seq. The DEGs were stratified into hormone-mediated signaling pathway, regulation of jasmonic acid-mediated signaling pathway, and response to abiotic stimulus (Supplementary Fig. 14a and Supplementary Data 2), indicating that the difference between Hap3 and Hap4 is involved in a diverse range of biological processes and molecular functions. Moreover, we conducted RNA-Seq analysis using WT, BnaA9.NF-YA7-OE, and BnaA9.NF-YA7-KO (aaCC, aacc, and AAcc homozygous mutant plants) seedlings after drought stress to explore the genome-wide transcriptional landscape dictated by *BnaA9.NF-YA7*. Through gene ontology (GO) analysis, these DEGs were found to be enriched in "response to water deprivation", "response to abscisic acid", "response to abiotic stimulus", and "response to osmotic stress" (Supplementary Fig. 14b, c and Supplementary Data 3, 4). Notably, Venn diagram analysis showed that aaCC and AAcc shared 4706 DEGs (Supplementary Fig. 14d), indicating that, to a great extent, *BnaA9.NF-YA7* and *BnaC5.NF-YA7* share similar downstream targets and plays largely redundant roles during these different biological processes.

However, we also found that the DEGs specifically regulated by *BnaA9.NF-YA7* or *BnaC5.NF-YA7* were stratified into different sets of biological processes (Supplementary Fig. 14e), suggesting that *BnaA9.NF-YA7* and *BnaC5.NF-YA7* may have distinct roles due to functional differentiation.

To better identify direct BnaA9.NF-YA7-binding motifs and target genes in *B. napus*, a ChIP-Seq assay was performed. A total of 4210 and 24,578 binding peaks were identified and associated with 3939 and 15,579 neighboring genes in the CK (normal conditions as control) and DS (drought stress) sets, respectively (Supplementary Fig. 15a and Supplementary Data 5). As expected, the canonical NF-Y-binding motif CCAAT was identified in the center of the ChIP-Seq peaks, but there were significant differences in the flanking sequence of the CCAAT motif identified between CK and DS (Supplementary Fig. 15b). The target genes of *BnaA9.NF-YA7* identified by ChIP-Seq were compared with the DEGs identified by RNA-Seq (Fig. 5a). In this way, common DEGs that were upregulated in BnaA9.NF-YA7-OE and downregulated in BnaA9.NF-YA7-KO or downregulated in BnaA9.NF-YA7-OE and

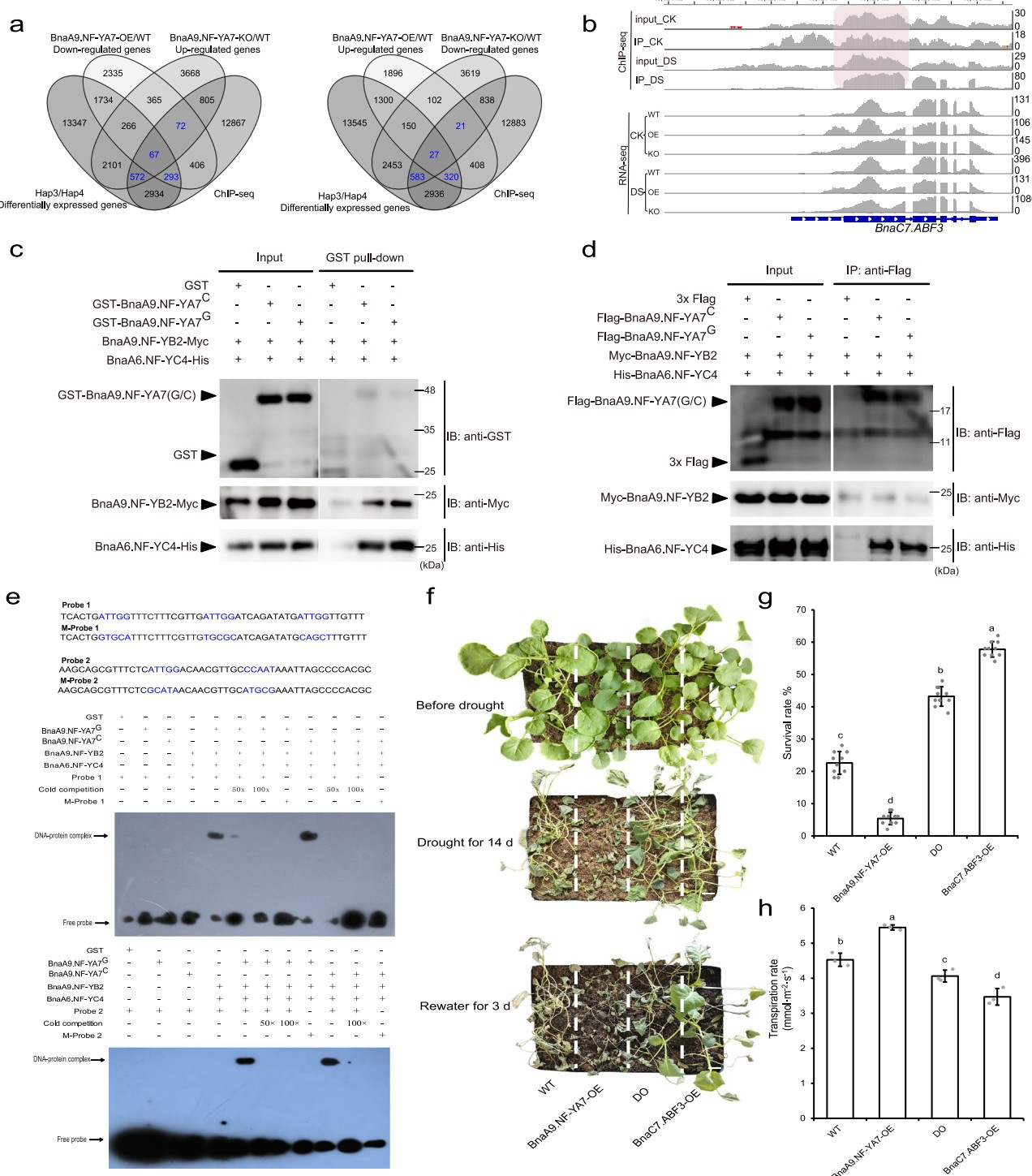

**Fig. 5 | *BnaA9.NF-YA7* directly targets *BnaABF3s*. a** Venn diagrams of the DEGs from transcriptomes and target genes of ChIP-Seq. **b** Genome browser view of normalized ChIP-Seq tags from the ChIP-Seq data and normalized RNA-Seq tags from the RNA-Seq data for *BnaA9.NF-YA7*, showing examples of *BnaC7.ABF3* bound by *BnaA9.NF-YA7* TFs. **c**, **d** GST pull-down (**c**) and co-immunicprecipitation assays (**d**) testing the interactions among BnaA9.NF-YA7 and BnaA9.NF-YB2/BnaA6.NF-YC4 in vitro and in vivo. **e** EMSA showing that NF-Y containing BnaA9.NF-YA7 could bind the CCAAT sites in the *BnaABF3s* promoter. Labeled probes containing the CCAAT element (Supplementary Fig. 18d, near TSS or within the coding region) were synthesized and labeled with biotin. The cold (unlabeled) probe and mutated CCAAT element (labeled) were used as a competitor and mutant probes,

respectively. **f** Phenotype of *B. napus* seedlings grown in plastic bins (length × width × depth = 32 × 20 × 10 cm) under well-watered, drought stress, or rewatering conditions. WT indicates wild-type ZS 11 plants, DO indicates *BnaA9.NF-YA7* and *BnaC7.ABF3* double overexpression lines Bar = 2 cm. **g** The survival rate of BnaA9.NF-YA7-OE, BnaC7.ABF3-OE, and double overexpression lines were compared with that of the WT. The values are means ± SD (*n* = 50 plants examined over ten independent experiments). **h** Transpiration rate of BnaA9.NF-YA7-OE, Bna-C7.ABF3-OE, and double overexpression lines were compared with WT under normal conditions. The values are means ± SD (*n* = 3 biological replicates). The lowercase letters indicate significant differences (*P* < 0.05, one-way ANOVA followed by a two-tailed LSD test). Source data are provided as a Source Data file.

upregulated in BnaA9.NF-YA7-KO were then extracted as high-confidence genes regulated by *BnaA9.NF-YA7*. A total of 1004 and 951 genes from the intersection of the Venn diagram were enriched in alpha-linolenic acid metabolism, biosynthesis of secondary metabolites, plant hormone signal transduction, and MAPK signaling pathway-plant (Supplementary Fig. 15c).

### *BnaA9.NF-YA7* directly targets *BnaABF3/4s*

Notably, the genes involved in plant hormone signal transduction included ABA-responsive element (ABRE)-binding factors (ABFs) (i.e., *BnaABF2s*, *BnaABF3s*, and *BnaABF4s*), which function redundantly as master regulators of ABA signaling in response to drought and osmotic stresses[20], and these genes were downregulated in the BnaA9.NF-YA7-OE lines, especially *BnaABF3s* and *BnaC1.ABF4* (Supplementary Data 6). *BnaAn.ABF3*, *BnaC1.ABF3*, *BnaC7.ABF3*, and *BnaC1.ABF4* were determined to have two to three potential binding sites containing CCAAT-box motifs in the promoter and coding regions (Fig. 5b and Supplementary Fig. 16a). In general, the ChIP–qPCR values were consistent with the ChIP-Seq results (Supplementary Fig. 16b). Y2H and BiFC assays were performed to identify the BnaNF-YB/YC dimer that can interact with BnaA9.NF-YA7 to form a trimer (Supplementary Fig. 17a, b), and each of these dimers were individually fused to GAL4 DNA-BD in a Y3H system to assess their potential interaction with the BnaA9.NF-YA7 protein (Supplementary Fig. 17c, d). Y3H, α-galactosidase activity, and FRET (fluorescence resonance energy transfer) assays showed that BnaA9.NF-YB2 and BnaA6.NF-YC4 could differentially form a trimer complex with BnaA9.NF-YA7 in Hap3 (representative *BnaA9.NF-YA7^G*) compared to that in Hap4 (representative *BnaA9.NF-YA7^C*) (Supplementary Fig. 18a–c). Both the GST pull-down and co-immunoprecipitation (Co-IP) assays showed that there were interactions among BnaA9.NF-YA7, BnaA9.NF-YB2, and BnaA6.NF-YC4 in vitro and in vivo (Fig. 5c, d). EMSA also revealed band shifts when BnaA9.NF-YA7, BnaA9.NF-YB2, and BnaA6.NF-YC4 were all present (Fig. 5e and Supplementary Fig. 18d). Interestingly, the dual-luciferase reporter assay showed that neither BnaA9.NF-YA7^G- nor BnaA9.NF-YA7^C-related trimers significantly induced the LUC activity of *Bna.ABF3/4* reporters (Supplementary Fig. 18e). Therefore, we speculated that *BnaA9.NF-YA7*-related trimers participate in *Bna.ABF3/4*-mediated regulatory pathways in an unknown way. To test this hypothesis, we examined the expression of downstream genes of *BnaABF3s* and *BnaC1.ABF4* in WT, BnaA9.NF-YA7-OE, and BnaA9.NF-YA7-KO. RT–qPCR analysis showed that the transcript levels of these downstream genes were significantly downregulated in BnaA9.NF-YA7-OE and slightly upregulated in BnaA9.NF-YA7-KO compared to WT under drought stress (Supplementary Fig. 19). These results suggest that *BnaA9.NF-YA7* represses the *BnaABF3/4*-regulated pathway through trimerization in response to drought stress.

Next, we further analyzed the downstream target gene *BnaC7.ABF3*, which is an upstream regulatory factor of *BnaA9.NF-YA7*. In comparison with the WT, overexpression of *BnaC7.ABF3* alone significantly increased survival because of increased leaf water holding capacity and decreased transpiration and water loss rates (Fig. 5f–h and Supplementary Fig. 20). In contrast, RNAi silencing of *BnaC7.ABF3*, led to a similar phenotype as that of the BnaA9.NF-YA7-OE lines that showed increased sensitivity to drought stress, while co-overexpression of *BnaA9.NF-YA7* and *BnaC7.ABF3* led to an intermediate phenotype (Fig. 5f and Supplementary Fig. 20). These results suggest that there may be a negative feedback regulatory loop between *BnaA9.NF-YA7* and *BnaC7.ABF3*.

### *BnaA9.NF-YA7* regulates drought responses via regulation of the H3K36me3 status

Previous studies have established a correlation between NF-Y binding and posttranslational modifications, particularly histone modifications[16,21–24]. To examine the specific methyltransferase

activities of *BnaA9.NF-YA7*, the histone modification status was detected using specific antibodies through western blotting, and H3K36me3 levels were found to be specifically increased in the knockout lines and reduced in the overexpression transgenic lines compared with those of the WT (Fig. 6a, b). RNA-Seq and RT-qPCR showed that BnaA9.NF-YA7 could repress *Bna.ASHH4s* expression, which encodes a histone-lysine N-methyltransferase involved in histone H3K36 methyltransferase activity (Fig. 6c). Correspondingly, ChIP analysis of H3K36 trimethylation at the *BnaC1.ABF3* and *BnaC7.ABF3* loci in different genotypes revealed that overexpression of *BnaA9.NF-YA7* significantly reduced the extent of H3K36 trimethylation at *BnaC1.ABF3* and *BnaC7.ABF3* (Fig. 6d, e). Comparable changes were not observed in chromatin at the *BnaA2.ACT7* locus, which was used as a negative control (Supplementary Fig. 21a). Next, we performed virus-induced *Bna.ASHH4s* silencing using a tobacco rattle virus (TRV)-based system to reduce *Bna.ASHH4s* mRNA levels and determined the expression level of *BnaC1.ABF3* and *BnaC7.ABF3* in TRV2-Bna.ASHH4 plants. The results indicated that *BnaC1.ABF3* and *BnaC7.ABF3* expression were positively correlated with *Bna.ASHH4s* expression. The expression of *BnaC1.ABF3* and *BnaC7.ABF3* in TRV2-Bna.ASHH4 plants did not significantly increase even under dehydration conditions (Supplementary Fig. 21b). In turn, H3K36 trimethylation at *BnaC1.ABF3* and *BnaC7.ABF3* showed significant reduction in the TRV2-Bna.ASHH4 lines (Supplementary Fig. 21c). The expression level of *BnaA9.NF-YA7* was not affected by the downregulation of *Bna.ASHH4s* under normal conditions. Moreover, ChIP analysis of H3K36 trimethylation in Hap3 and Hap4 genotypes revealed that high expression level of *BnaA9.NF-YA7* significantly reduced the extent of H3K36 trimethylation at *BnaC1.ABF3* and *BnaC7.ABF3* loci but not *BnaA4.DOR* and *BnaA2.ACT7* loci (Supplementary Fig. 22). Taken together, these results indicate that *BnaA9.NF-YA7* suppresses the expression of *BnaABF3s* by indirect regulation by H3K36me3 via *Bna.ASHH4s*.

### *BnaA9.NF-YA7* inhibits ABA-induced stomatal closure via upregulating *BnaA4.DOR* expression

Among these common DEGs, *BnaA4.DOR*, which encodes the F-box protein DOR that functions as an inhibitory factor for ABA-induced stomatal closure under drought stress[25,26], was significantly upregulated in the BnaA9.NF-YA7-OE lines but barely expressed in the other genotypes (Supplementary Fig. 19). ChIP-Seq and EMSA showed that BnaA9.NF-YA7 could bind the *BnaA4.DOR* promoter in vivo and in vitro (Fig. 7a, b). However, ChIP analysis of H3K36 trimethylation at the *BnaA4.DOR* locus in different genotypes showed no significant change in the extent of H3K36 trimethylation at *BnaA4.DOR* (Supplementary Figs. 21a and 22). The expression level of *BnaA4.DOR* in Hap3 was significantly higher than that in Hap4 (Fig. 7c). Transient expression assays confirmed that co-expression of BnaA9.NF-YA7^G with the NF-YB/C dimer induced higher LUC activity of the *pBnaA4.DOR* reporter than co-expression of BnaA9.NF-YA7^C with NF-YB/C dimer (Fig. 7d). To further determine the role of *BnaA4.DOR* in BnaA9.NF-YA7-OE mediated drought sensitivity, we performed virus-induced *BnaA4.DOR* silencing using a TRV-based system to reduce *BnaA4.DOR* mRNA levels in BnaA9.NF-YA7-OE leaves (Fig. 7e). Most TRV2-BnaA4.DOR plants survived after a subsequent 5-day recovery by watering. Although the survival rate was still lower than that of the WT, it was much higher than that of BnaA9.NF-YA7-OE plants (Fig. 7e, f). The water loss rate was significantly lower in BnaA4.DOR-silenced plants than in control plants (Fig. 7g). Furthermore, the percentage of closed stomata was significantly higher in TRV2-BnaA4.DOR plants (46.80%) compared to control plants (30.75%) (Fig. 7h) and exhibiting reduced stomatal conductance and a lower transpiration rate under nonstress conditions (Fig. 7i, j). These results may explain why overexpression of *BnaA9.NF-YA7* increases ABA sensitivity but does not significantly induce stomatal closure.

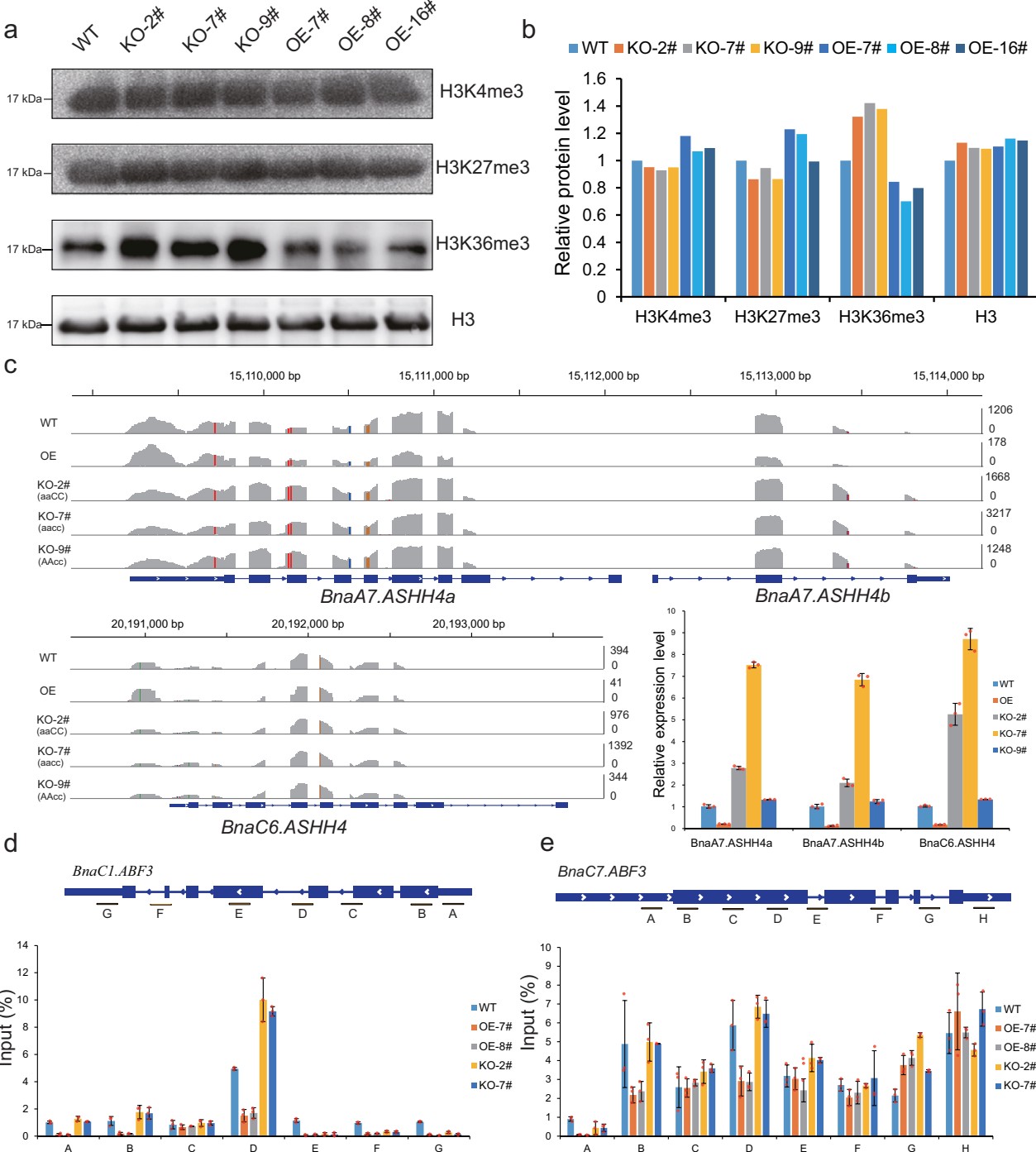

**Fig. 6 | BnaA9.NF-YA7 regulates drought responses via regulation of the histone H3 lysine 36 trimethylation (H3K36me3) status of the BnaABF3s locus.**
**a** Examination of H3K36 methyltransferase activity. Equal amounts of total proteins extracted from the nuclei of the wild-type (WT), *BnaA9.NF-YA7* knockout lines (KO-2#, KO-7#, and KO-9#), and *BnaA9.NF-YA7* overexpression lines (OE-7#, OE-8#, and OE-16#) were used for western blotting assays using different histone methylation antibodies, with H3 antibody as a loading control. **b** Relative protein levels in the WT, *BnaA9.NF-YA7* knockout lines, and *BnaA9.NF-YA7* overexpression lines. The amount of protein was determined by first normalizing the band intensity of

specific antibodies using ImageJ software. **c** Genome browser view of RNA-Seq tags from the RNA-Seq data and RT-qPCR show that *Bna.ASHH4s*, which encodes a histone-lysine N-methyltransferase involved in histone H3K36 methyltransferase activity, repressed by *BnaA9.NF-YA7*. The values are means ± SD (*n* = 3 biological replicates). **d, e** *BnaA9.NF-YA7* regulates H3K36me3 levels at the *BnaC1.ABF3* locus (**d**) and B*naC7.ABF3* locus (**e**) under drought stress. ChIP–qPCR was performed using an H3K36me3 antibody. The x-axis denotes different genetic regions of *BnaC1.ABF3* (**d**) and *BnaC7.ABF3* (**e**). The values are means ± SD (*n* = 3 biological replicates). Source data are provided as a Source Data file.

## Discussion

Natural variation in functional genes play extremely important roles in crop trait divergence. In rice, two SNPs in the coding region of NRT1.1B contribute to higher nitrogen-use efficiency and improved grain yield[27]. A tandem repeat of the CCATTC sequence in the

*OsSPL13* 5′ UTR causes reduced expression levels of *OsSPL13* and results in small grains[28]. In maize, a 366-bp insertion in the promoter of ZmVPP1 contributes significantly to drought tolerance[29]. A two-base sequence polymorphism in *UPA2* was associated with the different binding affinities of *DRL1*, and the wild-type *UPA2* allele

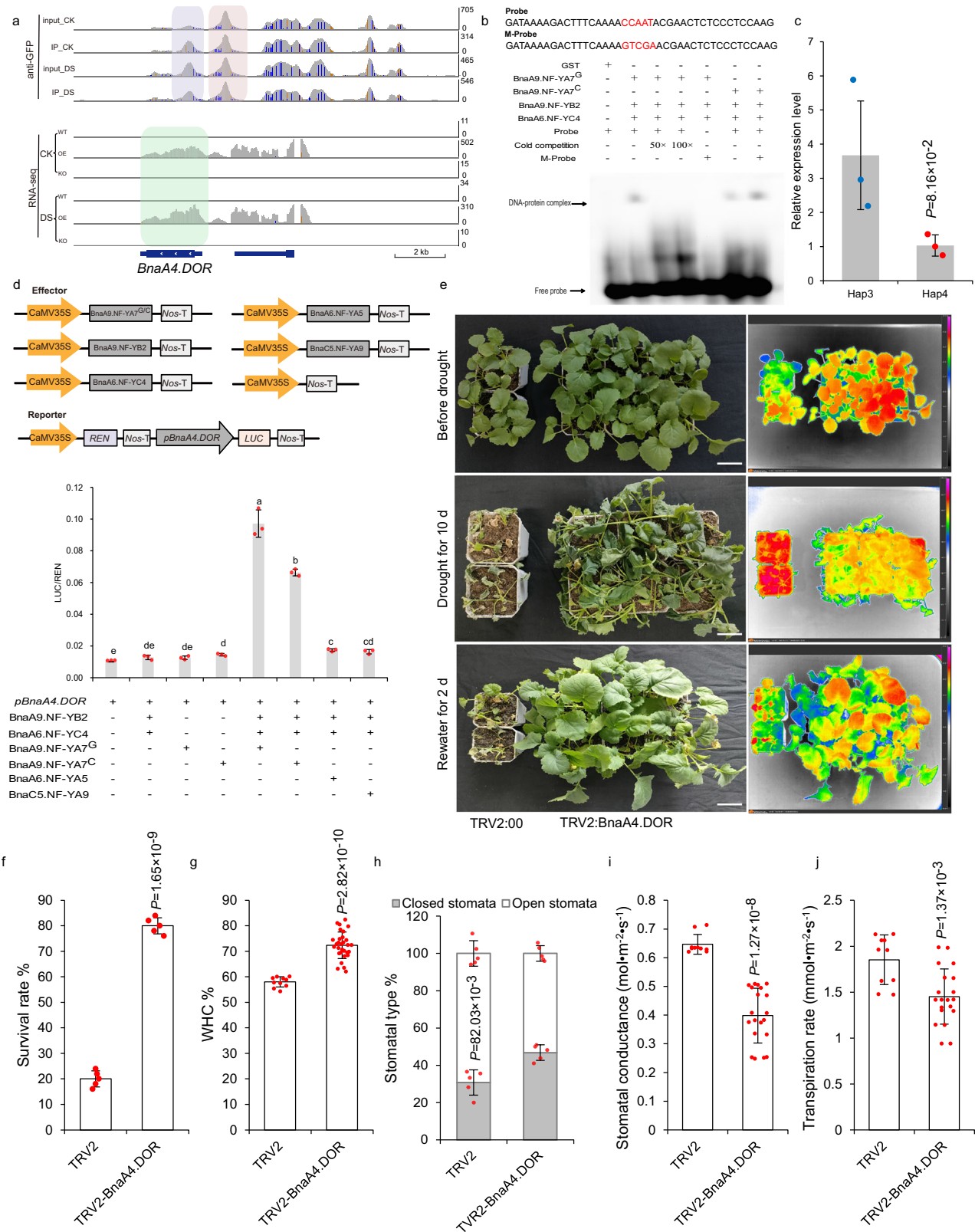

showed narrow leaf angles and promoted yield under high-density planting in maize[30]. Natural variation in *ZmFBL41* confers resistance to *Rhizoctonia solani*, that two amino acid substitutions prevent its interaction with *ZmCAD*, resulting in the inhibition of *ZmCAD* degradation and, consequently, the accumulation of lignin and the restriction of lesion expansion[31]. In cucumber, Che et al. identified a nonsynonymous polymorphism (G to A) in *CsCRC* as underlying the major effect of fruit size[32]. They found that *CsCRC^G*, but not *CsCRC^A*, targets the downstream auxin-responsive protein gene *CsARP1* to regulate its expression[32]. These reports have confirmed that natural variations and their favorable alleles can be utilized as important genetic resources for trait improvement in crop breeding. Here, we

**Fig. 7 | BnaA9.NF-YA7 induces BnaA4.DOR under drought stress, thereby increasing the rate of water loss. a** Genome browser view of normalized ChIP-seq tags from the ChIP-seq data and normalized RNA-seq tags from the RNA-seq data for *BnaA9.NF-YA7*, showing that *BnaA4.DOR* binds *BnaA9.NF-YA7*. **b** EMSA showing that NF-Y containing BnaA9.NF-YA7 could bind the CCAAT sites in the *BnaA4.DOR* promoter. **c** Expression levels of *BnaA4.DOR* in Hap3 and Hap4 under normal conditions. The values are means ± SD ($n = 3$ biological replicates). *P* values were calculated with the two-tailed Student's *t*-test. **d** Co-expression of Bna.NF-YAs with the NF-YB/C dimer activate the expression of *BnaA4.DOR*. The values are means ± SD ($n = 3$ biological replicates). The lowercase letters indicate significant differences ($P < 0.05$, one-way ANOVA followed by a two-tailed LSD test). **e** Phenotype of BnaA4.DOR-silenced BnaA9.NF-YA7-OE plants. Four-week-old TRV2: BnaA4.DOR and TRV2:00 plants were exposed to drought stress by withholding watering for 10 days and rewatering for 2 days. Representative images and infrared thermal images were taken before and after drought stress and rewatering. White bars = 5 cm. **f** Survival rates of plants after rewatering. The values are means ± SDs ($n = 50$ plants examined over five independent experiments). *P* values were calculated with the two-tailed Student's *t*-test. **g** The water holding capacity of TRV2: BnaA4.DOR and TRV2:00 leaves. The values are means ± SD ($n = 10$ biologically independent plants in TRV2; $n = 30$ biologically independent plants in TRV2: BnaA4.DOR). *P* values were calculated with the two-tailed Student's *t*-test. **h** Stomata type (%) in TRV2: BnaA4.DOR and TRV2:00 leaves under normal conditions. The values are means ± SD ($n = 10$ biologically independent experiments). *P* values were calculated with the two-tailed Student's *t*-test. **i, j** Stomatal conductance (**i**) and transpiration rate (**j**) in TRV2: BnaA4.DOR and TRV2:00 leaves under normal conditions. The values are means ± SD ($n = 10$ biological replicates in TRV2; $n = 20$ biological replicates in TRV2: BnaA4.DOR). *P* values were calculated with the two-tailed Student's *t*-test. Source data are provided as a Source Data file.

found that a nonsynonymous polymorphism (G to C, M63I) and deletion of the CCAAT-box of the promoter in *BnaA9.NF-YA7* affects the drought tolerance of *B. napus*. The accessions with the *BnaA9.NF-YA7$^C$* allele had a higher WHC than the accessions with the *BnaA9.NF-YA7$^G$* allele under drought treatment. The materials carrying the C allele were further divided into two types (Hap1 and Hap4), and Hap4 had stronger drought tolerance than Hap1 (Supplementary Fig. 5c). However, deletion of the CCAAT-box, which reduced the *BnaA9.NF-YA7* expression level, occurred only in Hap4, suggesting that the M63I substitution cooperates with CCAAT-box deletion and can be used to increase the drought tolerance of *B. napus*.

Crops, including *B. napus*, exist in constantly changing environments and are often exposed to large amounts of abiotic and biotic stresses during their lifecycle. Among these stresses, drought stress is one of major environmental factors affecting the geographical distribution of plants in nature, limiting crop productivity in agriculture, and threatening food security[33]. The ABA signaling pathway is central to drought- and salt-stress responses in plants[34]. ABFs are known as master regulators of ABA signaling in response to drought and osmotic stresses, and the activation of genes containing an ABA-responsive element (ABRE) is largely dependent on ABF transcription factors[20,35,36]. In the presence of ABA, the activation of SnRK2s leads to the phosphorylation of ABF proteins, thus allowing ABFs to induce the expression of other ABA-responsive genes[20,37]. In this study, we found that fluridone (FLU) inhibited the upregulated expression of *BnaA9.NF-YA7* induced by ABA treatment (Supplementary Fig. 5f), and ABFs directly regulated *BnaA9.NF-YA7* expression (Fig. 4), suggesting that *BnaA9.NF-YA7* may also mediate the regulation of the drought response via an ABA-dependent pathway. Under stress conditions, the activation of ABA signaling helps plants respond to environmental stress and thus help them survive under adverse conditions. However, overexpression of drought-responsive genes often results in growth deficits and yield loss[38]. Therefore, balancing ABA signaling is very important for plants' ability to cope with the changing environment[39]. Studies on switching off ABA signaling have mostly focused on the genes of the PYL-PP2C-SnRK2 pathway, which is the core pathway of ABA signal transduction, and few reports have focused on ABFs and their downstream genes. In this study, we found that the NF-Y complex involving BnaA9.NF-YA7 specifically binds to the promoter and CDS of *BnaABF3s* and reduces histone H3 Lys 36 (H3K36) trimethylation levels to inhibit their expression, thus attenuating ABA signaling (Figs. 5 and 6). These results suggest that the negative feedback regulatory network mediated by *BnaA9.NF-YA7* serves as a supplementary mechanism for ABA signal balance.

Stomatal closure preserves water in the plant, and ABA has a role in the regulation of stomatal movement to prevent water loss[40]. Because stomatal movements control $CO_2$ influx and transpiration, drought tolerance strategies to reduce water loss via stomatal closure occur at the expense of photosynthesis, growth, and yield[38]. Here, we found that overexpression of *BnaA9.NF-YA7* increased ABA

sensitivity, but the percentage of closed stomata was not significantly affected by ABA (Fig 2l-q). We propose that this strategy may be part of an ABA signaling balance mechanism. *BnaA9.NF-YA7* and its NF-Y complexes directly interact with the *BnaA4.DOR* promoter upregulates its expression level, thereby maintaining partial stomatal opening under drought stress and ensuring the $CO_2$ influx needed for plant photosynthesis, allowing plants to continue to grow under drought stress conditions. The M63I substitution occurred in a Pro-rich transcriptional activation domain, resulting in a transcriptional activation difference between *BnaA9.NF-YA7$^G$* and *BnaA9.NF-YA7$^C$* (Fig. 7d), thus breaking the fine-tuning of the balance between opening and closing stomata under drought stress. Nevertheless, the impact of the M63I mutation on drought tolerance is limited and only when accompanied by the CCAAT mutation can it further enhance drought tolerance. This may also explain why Hap4 exhibits stronger drought tolerance than Hap1 (Fig. 1c, d). Therefore, we posit that these two mutations are both independent yet intricately linked, with CCAAT primarily influencing the feedback regulation loop, while M63I predominantly modulates stoma sensitivity toward ABA. The excessive expression of *BnaA9.NF-YA7$^{G/C}$* excessively inhibited the transmission of ABA signals, and although the overexpression of *BnaA9.NF-YA7$^C$* promotes stomatal closure, it is insufficient to reverse the drought sensitivity caused by the inhibition of ABA signaling (Fig. 2f). Taken together, the data indicate that *BnaA9.NF-YA7* plays a dual role as both an inhibitor and activator, effectively regulating ABA signaling in rapeseed to maintain a balance between growth and drought stress responses.

In summary, we propose the following hypothesis for how the drought tolerance difference between Hap4 and other haplotypes is regulated (Fig. 8): the expression of *ABF* genes and the accumulation of ABF proteins are induced by ABA signaling under drought stress, thus allowing ABFs to induce the expression of *BnaA9.NF-YA7* and other ABA-responsive genes. The upregulated *BnaA9.NF-YA7* can regulate the cascade strength of ABA signaling by reducing the H3K36me3 level to indirectly inhibit the expression of *BnaABF3/4s*, and thus *BnaABF3/4s* and *BnaA9.NF-YA7* forms a regulatory loop to prevent excessive amplification of ABA signaling, thereby maintaining the dynamic balance between growth and the stress response in *B. napus*. Compared with the other haplotypes, Hap4, containing a two-base sequence variant (CCAAT-box deletion) at the promoter, is associated with a lower binding affinity for NF-Y. Low *BnaA9.NF-YA7* expression in Hap4 and differences in transcriptional activation produced by the M63I substitution result in higher *BnaABF3/4s* levels and a lower *BnaA4.DOR* level, consequently decreasing the transpiration rate and water loss rate and increasing the leaf water holding capacity. Although the contribution of *BnaA9.NF-YA7* alone to improve *B. napus* drought tolerance was limited, our findings provide prospective drought tolerance allele. Furthermore, we add insights into the mechanisms that control ABA signaling homeostasis.

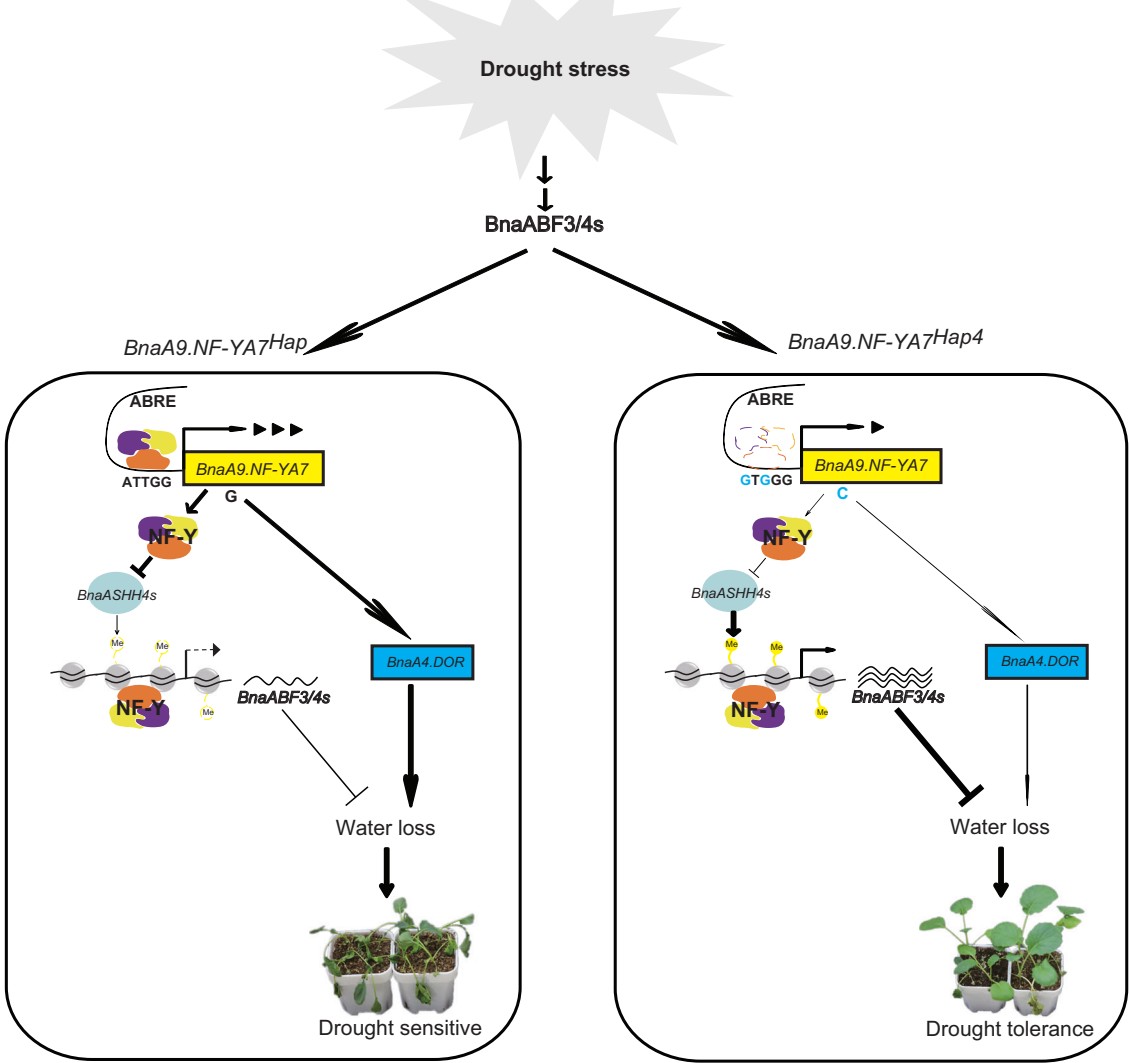

**Fig. 8 | Working model demonstrating how the drought tolerance difference between Hap4 and other haplotypes is regulated at the molecular level.** All haplotypes except Hap4 are collectively referred to as *BnaA9.NF-YA7^Hap^*. Arrows indicate upregulation, and lines ending with bars represent downregulation. The thickness of the lines represents the strength of the regulation. The BnaABF3/4 proteins promote the upregulation of *BnaA9.NF-YA7* expression by direct DNA-binding at its promoter region under drought stress. To prevent excessive amplification of the ABA signal, the upregulated *BnaA9.NF-YA7* regulates the cascade strength of the ABA signal by reducing the H3K36me3 level to inhibit the expression of *BnaABF3/4s*. On the other hand, *BnaA9.NF-YA7* negatively regulates drought tolerance by promoting *BnaA4.DOR* activity via direct DNA binding at its promoter region. Hap4 containing a two-base sequence variant (CCAAT-box deletion) at the promoter is associated with a lower binding affinity for NF-Y than the other haplotypes. Low *BnaA9.NF-YA7* expression in Hap4 and differences in transcriptional activation produced by the M63I substitution result in high *BnaABF3/4s* levels and low *BnaA4.DOR* level, consequently decreasing the transpiration rate and water loss rate and leading to stronger drought resistance.

## Methods

### Plant materials
The diversity panel used in this study was from a previous report[18]. The *B. napus* transgenic lines used in this study were in the Zhongshuang 11 (ZS 11, Hap3) genetic background. The *ProBnaA9.NF-YA7*:GUS transgenic line harboring a 1697 bp *BnaA9.NF-YA7* promoter fragment was transformed into *Arabidopsis* (Col-0) by floral dipping.

### Association analysis
A total of 205 overlapping lines from previous reports of Tan et al. and Lu et al. were used for association analysis of WHC[17,18]. The GWAS was conducted using the mixed linear model (MLM)[41] with a total of 581,532 SNPs (miss data <20%, MAF >0.05) in the GAPIT R package[42]. The Manhattan plot was displayed using the CMplot R package[43].

### Phylogenetic analysis of *BnaA9.NF-YA7*
The full-length amino acid sequence of *BnaA9.NF-YA7* was used for a BLAST search of the NCBI database (http://www.ncbi.nlm.nih.gov/) to identify homologous genes in *Brassicae* species. The amino acid sequences were aligned using ClustalX 2.1 software. A phylogenetic tree was constructed using the neighbor-joining method with bootstrapping (1000 replicates) in MEGA version 5[44].

### Plasmid construction and *B. napus* transformation
The *BnaA9.NF-YA7* overexpression construct was generated with pENTR™ Directional TOPO® Cloning Kits (Invitrogen, USA) according to the manufacturer's protocols, and the relevant fragment was cloned and inserted into the binary vector pEarleyGate 101. For *BnaC7.ABF3* overexpression, the full-length CDS of *BnaC7.ABF3* without the stop codon was amplified and cloned between the *Bam*HI and *Sal*I enzyme

sites of the expression vector to generate the 35S:BnaC7.ABF3-DsRed fusion plasmid. The recombinant plasmids were introduced into the *B. napus* cultivar ZS 11 through *Agrobacterium*-mediated transformation. The transgenic plants were verified by PCR, and the expression levels in the transgenic plants were confirmed by RT-qPCR.

For *BnaA9.NF-YA7* knockout, two 18-bp targeting sequences were selected within the *BnaA9.NF-YA7* and *BnaC5.NF-YA7* gene. The sequences were synthesized and inserted into the p2×35S-dpCas9-atU6-Grna-35S-Hy vector to produce the CRISPR–Cas9 plasmid. The CRISPR–Cas9 plasmid was introduced into the *B. napus* cultivar ZS 11 through Agrobacterium-mediated transformation. The target was amplified by PCR using primer pairs flanking the designed target site, and the PCR product was sequenced.

For the RNA interference (RNAi) construct, a unique 187-bp fragment from the cDNA of *BnaA9.NF-YA7* was amplified and cloned between the *Bam*HI and *Xba*I enzyme sites of the pFGC5941M vector, and the antisense fragment was cloned into between the *Nco*I and *Aat*II enzyme sites of the pFGC5941M vector to form a hairpin structure. A unique 191-bp fragment from the cDNA of *BnaC7.ABF3* was amplified and cloned into the pFGC5941M vector. The recombinant plasmids were introduced into the *B. napus* cultivar ZS 11 through *Agrobacterium*-mediated transformation. The transgenic plants were verified by PCR, and the expression levels in the transgenic plants were confirmed by RT-qPCR.

For VIGS (virus-induced gene silencing), a unique 331-bp fragment from the cDNA of *BnaA4.DOR* and a unique 212-bp fragment from the cDNA of *Bna.ASHH4* were amplified and reverse-cloned between the *Bam*HI and *Eco*RI enzyme sites of the pTRV2 vector, respectively. pTRV1 and pTRV2 were transformed into *Agrobacterium tumefaciens* (strain GV3101) and subsequently used for the infection of *B. napus* seedlings.

For site-directed mutagenesis, the site-directed mutation of the 189th base of Hap3 (G to C) was achieved by overlap extension PCR, followed by cloning the relevant fragment into the binary vector pCAMBIA1302 to create *pBnaA9.NF-YA7^H3*:Hap3, *pBnaA9.NF-YA7^H4*:Hap4, and *pBnaA9.NF-YA7^H3*:Hap3^C constructs, respectively. Transgenic *Arabidopsis* plants were generated through floral-dip by *Agrobacterium*-mediated transformation.

## Stress treatment and physiological experiments

To assess the drought stress tolerance of WT and transgenic plants, 7-day-old seedlings were transferred to pots, grown for 3 weeks under normal conditions, and subjected to drought conditions by withholding water from the seedlings for 1 week, 10 days, or 14 days[45]. The drought-related phenotypes were photographed with a camera (DS126611, Canon). The leaf water loss rate and relative water content (RWC) were determined as previously described by refs. [8,46]. Briefly, at least five leaves from different plants of each genotype were detached from 3-week-old plants and placed in a petri dish on a laboratory bench ($22 \pm 1\,°C$ and 30–40% RH), and the weight of the detached leaves was then measured every 1 h for a period of 48 or 12 h. The experiment was repeated three times. Water loss was presented as the percentage of initial water loss. For the RWC, the leaves were detached from 3-week-old plants and weighed to obtain leaves weight (W), after which the leaves are immediately hydrated to full turgidity for 12 h under normal room light and temperature. After hydration, the leaves are weighed to obtain fully turgid weight (TW). Leaves are then oven-dried at 85 °C for 24 h and weighed to determine dry weight (DW). The relative water content was calculated using the following formula:

$$RWC(\%) = [(W - DW)/(TW - DW)] \times 100 \qquad (1)$$

To measure water holding capacity (WHC), fully expanded fourth leaves were detached from the seedlings, and the petioles were wrapped with fresh-keeping film to prevent water loss from the petioles. The fresh weight of detachment leaves was measured immediately detached from the plants ($W_f$), 24 h dark treatment ($23 \pm 1\,°C$ and 30–40% RH, $W_t$), and drying ($W_d$). The water holding capacity of separated leaves was calculated using the following formula shown as percentage:

$$WHC(\%) = (W_t - W_d)/(W_f - W_d) \times 100 \qquad (2)$$

The experiment was repeated three times. For each repeat, at least five leaves from different plants of each genotype were used. The soluble protein content was determined by the Coomassie Brilliant Blue G250 method, and the free proline content was measured by the sulfosalicylic acid colorimetric method[47,48]. For the stomatal aperture assay, epidermal strips were peeled from 4-week-old plants, and the mesophyll cells were removed with a small brush[45]. For the ABA-mediated stomatal closure assay, the leaves were incubated in MES buffer (10 mM MES-KOH, pH 6.15, 10 mM KCl, and 50 mM $CaCl_2$) in light for 3 h at 22 °C to open the stomata. After the addition of 20 mM ABA or DMSO (control) to the buffer and incubation of the leaves for an additional 2 h, images were taken by an optical microscope equipped with a digital camera (Nikon ECLIPSE E600, Japan). The stomatal aperture was analyzed using ImageJ software, and three independent repeats were performed.

## Histochemical GUS staining and GUS activity

The promoter sequence of *BnaA9.NF-YA7* was amplified from ZS 11 (Hap3) and R219 (Hap4) and inserted into the binary vector pCAMBIA1305.1 to create *pBnaA9.NF-YA7^Hap3*:GUS and *pBnaA9.NF-YA7^Hap4*:GUS reporter gene constructs, respectively. For promoter truncation, differently truncated promoters of *BnaA9.NF-YA7* were amplified and cloned between the *Xba*I and *Nco*I enzyme sites of the pCAMBIA1305.1 vector to generate *promoter*:GUS fusion plasmids. The recombinant plasmids were transformed into *Arabidopsis* (Col-0) by floral dipping. For the positive lines, plants from the T3 generation were used for GUS staining, which was performed according to the method described in the manufacturer's protocols for the GUS Staining Kit (Coolaber, China). The stained tissues were photographed by an optical microscope equipped with a digital camera (Nikon SMZ1500, Japan). GUS activity was measured by the fluorometry method[49].

## RNA extraction, qRT–PCR, and RNA-sequencing

Total RNA was extracted from 4-week-old seedlings before and after drought treatment with an EZ-10 DNAaway RNA Mini-Prep Kit (Sangon Biotech, China) according to the manufacturer's protocols. For RT–qPCR, cDNA was synthesized from the total RNA using a PrimeScript™ RT reagent kit with gDNA Eraser (TaKaRa, Japan) as described in the supplier's instructions. RT–qPCR analyses were performed with TB Green® Premix Ex Taq™ II (Takara) on a CFX96 Real-Time PCR Detection System. PCR amplification was performed with a 3 min denaturing step at 95 °C, followed by 40 cycles of 95 °C for 15 s and 60 °C for 30 s. Three independent biological replicates were collected for each sample. *Bna.Actin7* was used as the internal control. The comparative CT ($2^{-\Delta CT}$) method[50] was used to quantify the relative gene expression levels.

For RNA-Seq, total RNA was extracted from *B. napus* leaves of Hap3, Hap4, WT, BnaA9.NF-YA7-OE, and BnaA9.NF-YA7-KO lines before and after drought treatment. High-quality RNA was sent to Biomarker Technologies Co., Ltd. (Beijing, China) for sequencing and analysis. TopHat2 software was used for reference genome (Darmor-bzh v4.1) mapping. Visualize the alignment results of Reads on the genome using IGV software. Transcript abundance was calculated as fragments per kilobase of transcript per million mapped reads (FPKM) values. Differentially expressed genes (DEGs) of two conditions/groups was performed using the DESeq2. The resulting *P* values were adjusted using Benjamini and Hochberg's approach for controlling the

false discovery rate. Genes with an adjusted *P*-value < 0.05 found by DESeq2 were assigned as differentially expressed. The FDR <0.05 and fold change ≥1.5 was set as the threshold for significantly differential expression.

## Thermal imaging

Thermal imaging of plants was performed as described previously with some modifications[45]. Briefly, the plants were grown at 20 to 22 °C under a 16-h-light/8-h-dark cycle in a greenhouse for 4 weeks. Water was then withheld for 1 week before measurements. Thermal images were obtained using a WIC640 infrared camera (Workswell, Czech Republic). Leaf temperature was measured using Workswell Core-Player professional software. For each genotype, plants in three different pots were analyzed, with similar results.

## Yeast one-hybrid, two-hybrid, and three-hybrid assays

An approximately 0.5 kb core promoter DNA fragment of *BnaA9.NF-YA7* was inserted upstream of the *AbA^r* reporter gene in the pBait-AbAi vector as the pBait-AbAi plasmid. A *B. napus* cDNA library was prepared in the pGADT7-Rec vector in the Matchmaker® Gold Yeast One-Hybrid Library Screening System (PT4087-1, Clontech, USA) using RNA isolated from 4-week-old seedlings after drought stress treatment according to the manufacturer's instructions. Yeast one-hybrid screening and one-hybrid interaction assays were performed by the user manual supplied with the Matchmaker Gold Yeast One-Hybrid Library Screening System (PT4087-1, Clontech, USA).

For the yeast two-hybrid assay (Y2H), the full-length *BnaA9.NF-YA7* coding sequence was cloned and inserted into the *Eco*RI and *Bam*HI sites of the prey vector pGADT7 or into the *Eco*RI and *Bam*HI sites of the bait vector pGBKT7. The full-length CDSs of *BnaNF-YB* genes and *BnaNF-YC* genes were cloned and inserted into the pGADT7 and pGBKT7 vectors, respectively. The coding region of *BnaC7.ABF3* was amplified and inserted into pGADT7 as AD-BnaC7.ABF3 and *BnaA3.ABF4* was amplified and inserted into pGBKT7 as BD-BnaA3.ABF4. For interaction assays, the specific bait and prey constructs were co-transformed into yeast Y2HGold cells using the Yeastmaker™ Yeast Transformation System 2 (PT1172-1, Clontech, USA) according to the manufacturer's instructions and grown on SD/-Trp/-Leu and SD/Trp/-Leu/-His/-Ade media for interaction tests.

For the yeast three-hybrid assay (Y3H), the pBridge vector was used to express both BnaNF-YBs and BnaNF-YCs. The pGADT7 vector was used to express BnaNF-YAs, *BnaC7.ABF3*, and *BnaA3.ABF4*. Moreover, the *BnaA9.NF-YB2* coding sequence was cloned and inserted into the *Eco*RI and *Bam*HI sites of multiple cloning site I (MCSI) and the *BnaA6.NF-YC4* coding sequence was cloned and inserted into the *Not*I and *Bgl*II sites of MCSII of the pBridge vector as a bait plasmid. A *B. napus* cDNA library was prepared in the pGADT7-Rec vector in the Matchmaker® Gold Yeast Two-Hybrid Library Screening System (PT4084-1, Clontech, USA) using RNA isolated from 4-week-old seedlings after drought stress treatment according to the manufacturer's instructions. Yeast three-hybrid screening and three-hybrid interaction assays were performed by the user manual supplied with the Matchmaker® Gold Yeast Two-Hybrid Library Screening System (PT4084-1, Clontech, USA).

## BiFC assay

The coding regions of BnaNF-Ys were amplified and inserted into the pXY104 vector containing the C-terminus of yellow fluorescent protein (cYFP) and the pXY106 vector containing the N-terminus of yellow fluorescent protein (nYFP). The coding region of *BnaC7.ABF3* was amplified and inserted into the pXY104 vector and the coding region of *BnaA3.ABF4* was amplified and inserted into the pXY106 vector. The plasmids were then transformed into *Agrobacterium* GV3101 cells with the silencing suppressor P19 strain. Then, these strains were resuspended in an induction medium (10 mM MES-KOH, pH 5.7, 10 mM

MgCl₂, 200 mM acetosyringone) to an OD₆₀₀ of 1.0. After incubation at room temperature for 3 h in darkness, the different strain combinations were mixed and infiltrated into *Nicotiana benthamiana* leaves. The plants were grown at 22 °C under a 12-h-light/12-h-dark cycle in a greenhouse for 48 to 72 h, and YFP signals were observed under a confocal laser-scanning microscope (Zeiss LSM 800).

## BiFC-FRET analysis

For BiFC-based FRET analysis, the coding regions of *BnaA9.NF-YB2* and *BnaA6.NF-YC4* were cloned and inserted into the pXY106 vector and pXY104 vector to produce nYFP-BnaA9.NF-YB2 and cYFP-BnaA6.NF-YC4, respectively. The full-length sequences of *BnaA9.NF-YA7^G* and *BnaA9.NF-YA7^C* were amplified, and the N-terminus was fused to the CFP coding sequence, generating BnaA9.NF-YA7^G-CFP and BnaA9.NF-YA7^C-CFP, respectively. The plasmids were then transformed into *Agrobacteria* GV3101 cells with the silencing suppressor P19 strain. Then, *Agrobacteria* cultures containing equal amounts of nYFP-BnaA9.NF-YB2 and cYFP-BnaA6.NF-YC4 strains were mixed and infiltrated into *Nicotiana benthamiana* leaves to allow the formation of BiFC complexes (acceptors), along with equal amounts of BnaA9.NF-YA7^G-CFP or BnaA9.NF-YA7^C-CFP (donors) strain cultures. After infiltration, the plants were grown at 22 °C under a 12-h light/12-h dark cycle in a greenhouse for 48 to 72 h before analyses of interaction using FRET. An excitation wavelength of 458 nm and an emission wavelength of 470–500 nm was used for CFP, and an excitation wavelength of 514 nm and an emission wavelength of 515–545 nm were used for YFP. The FRET energy transfer efficiency was calculated using the following formula shown as percentage[51,52]:

$$E = (\text{donor after} - \text{donor before}) \times 100/\text{donor after} \qquad (3)$$

## Pull-down assay

The coding regions of *BnaA9.NF-YB2-Myc*, *BnaA6.NF-YC4-His*, *BnaC7.ABF3*, *BnaA3.ABF4*, *BnaA9.NF-YA7^G*, and *BnaA9.NF-YA7^C* were cloned and inserted into the pGEX-4T-1 vector to produce GST-BnaA9.NF-YB2-Myc, GST-BnaA6.NF-YC4-His, GST-BnaC7.ABF3, GST-BnaA3.ABF4, GST-BnaA9.NF-YA7^G, and GST-BnaA9.NF-YA7^C, respectively. The expression of GST recombinant proteins in *E. coli* Rosetta cells (DE3; TransGen Biotech, Beijing, China) was induced by treatment with 0.5 mM IPTG for 12 h. The proteins were purified using glutathione agarose beads (Beyotime Biotechnology, Beijing, China). The BnaA9.NF-YB2-Myc and BnaA6.NF-YC4-His proteins were obtained after the GST label was digested by thrombin following the manufacturer's instructions.

For the pull-down assay, equal amounts of BnaA9.NF-YB2-Myc and BnaA6.NF-YC4-His were incubated in binding buffer (50 mM Tris-HCl, pH 8.0, 100 mM NaCl, and 1 mM EDTA) with immobilized GST or GST-labeled proteins at 4 °C for 4 h. The Glutathione agarose beads were then mixed with the protein mixture in 1 mL of PBS (140 mM NaCl, 2.7 mM KCl, 10 mM Na₂HPO₄, and 1.8 mM KH₂PO₄) and mixed by rotation at 4 °C for 1 h. After washing three times with PBS, proteins that were retained on the beads were separated by SDS–polyacrylamide gel electrophoresis (PAGE). Western blot analysis was performed using anti-GST (1:2000), anti-Myc (1: 2000), and anti-His (1:2000) antibodies (Beyotime Biotechnology, Beijing, China).

## Co-IP assay

The recombinant vectors (containing Myc-BnaA9.NF-YB2, His-BnaA6.NF-YC4, Flag-BnaC7.ABF3, Flag-BnaA3.ABF4, Flag-BnaA9.NF-YA7^G, and Flag-BnaA9.NF-YA7^C, respectively) were transformed into *Agrobacterium tumefaciens* strain GV3101 carrying p19 (35 S:p19). Then, these strains were resuspended in an induction medium to an OD₆₀₀ of 0.8. After incubation at room temperature for 3 h in darkness, the different strain combinations were mixed and infiltrated into *N.*

*benthamiana* leaves. Total proteins were extracted from the samples with extraction buffer (50 mM Tris-MES (pH 8.0), 0.5 M sucrose, 1 mM MgCl₂, 10 mM EDTA, 5 mM DTT, 1 mM PMSF, and 100× protease inhibitor cocktail) at 3 d.p.i and incubated with anti-Flag antibody immobilized onto Protein G/A-Agarose suspension (Beyotime Biotechnology, Beijing, China) in extraction buffer at 4 °C for 4 h. After incubation, the beads were washed with wash buffer (100 mM Tris-HCl (pH 8.0), 100 mM NaCl, and 1% [v/v] Triton X-100) and eluted with 5× SDS loading buffer. The proteins bound to the beads were resolved by SDS–PAGE and detected by anti-FLAG, anti-Myc, and anti-His antibodies (Beyotime Biotechnology, Beijing, China).

### Transient expression assay

For the dual-luciferase transcriptional activity assays, the firefly luciferase (LUC) gene driven by the minimal TATA box of the CaMV 35S promoter following five copies of the GAL4 binding element was used as a reporter[53]. *BnaA9.NF-YA7* CDS fragments of different lengths fused with the yeast GAL4 DNA-binding domain driven by the CaMV 35S promoter followed by the translational enhancer from tobacco mosaic virus were used as effectors[54].

For the dual-luciferase transient expression assays, the different promoter segments were recombined into the *Hin*dIII and *Bam*HI sites of the pGreenII 0800-LUC vector, generating the *promoter*:LUC plasmids as reporters. The Renilla luciferase (REN) gene driven by the 35S promoter in the pGreenII 0800-LUC vector was used as the internal control[30]. The full-length CDSs of different genes were amplified and recombined into the *Bam*HI and *Eco*RI sites of the pGreenII 62-SK vector driven by the 35S promoter, forming different effectors. The empty pGreenII 62-SK vector was used as a control.

Transient dual-luciferase assays were performed in *Arabidopsis* protoplasts collected from the leaves of 4-week-old seedlings. The luciferase signal was detected using the Dual-Luciferase Reporter Assay System (Promega, USA) with a GloMax 20/20 Luminometer (Promega, USA). Relative LUC activity was calculated by normalizing LUC activity to REN activity[30].

### Chromatin immunoprecipitation assay

Chromatin immunoprecipitation (ChIP) assays were performed using an EpiQuik™ Plant ChIP kit (Epigentek) according to the user guide. Leaf tissues (1 g) before and after drought treatment were collected from 35S:*BnaA9.NF-YA7*-GFP transgenic seedlings grown in pots for 4 weeks. An anti-GFP antibody (ab290; Abcam) was used to immunoprecipitate the sonicated DNA–protein complexes containing *BnaA9.NF-YA7*-GFP. The qualified chromatin fragments were sent to Novogene (www.novogene.com/) for ChIP-Seq analysis. High-quality clean reads were mapped to the public *B. napus* genome Darmor-bzh v4.1 with default parameters using the Burrows–Wheeler Aligner (BWA)[55].

To determine whether *BnaABF3/4* regulates and controls *BnaA9.NF-YA7*, chromatin immunoprecipitation was performed using *BnaC7.ABF3-Flag*, *BnaA3.ABF4-Flag*, and empty vector transgenic plant leaves. Sonicated chromatin fragments were immunoprecipitated using an anti-Flag antibody (CST, 14793S). For histone modification analysis, ChIP experiments were performed using WT, BnaA9.NF-YA7-OE, and BnaA9.NF-YA7-KO leaves. ChIP was performed using Protein A/G magnetic beads (ab193262; Abcam) and anti-H3K36me3 antibody (Abcam, ab9050), anti-H3K4me3 antibody (Abcam, ab8580), anti-H3K27me3 antibody (Abcam, ab6002), and anti-Histone H3 antibody (Abcam, ab1791). The precipitated DNA fragments were recovered and analyzed by RT–qPCR using specific primers. ChIP–qPCR assays were performed using three biological replicates and three technical replicates for each biological replicate. The fold enrichment was calculated against the GFP or Flag control samples.

### Reporting summary

Further information on research design is available in the Nature Portfolio Reporting Summary linked to this article.

## Data availability

Sequencing data associated with this study have been deposited in the NCBI Sequence Read Archive under BioProject accessions PRJNA977421, PRJNA978264, and PRJNA978979. The primers used in this work are listed in Supplementary Table 2, Supplementary Data 7, and Supplementary Data 8. Source data are provided with this paper.

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

## Acknowledgements

We thank Huanhuan Jiang for providing the RNA interference vector, Xiupeng Mei (State Cultivation Base of Crop Stress Biology for Southern Mountainous Land, Southwest University) for providing the dual-luciferase transcriptional activity vector, and Lisha Zhang (Key Laboratory of Application and Safety Control of Genetically Modified Crops, Southwest University) for transient expression assays. This research was supported by the National Key Research and Development Program of China to L.L. (2022YFD1200400) and the National Natural Science Foundation of China to L.L. (31971902 and 32001509).

## Author contributions

L.L. designed the study. J.W. and L.M. performed the experiments. J.W., L.M., and Y.L. analyzed the data. C.Q., K.L., Z.T., J.L., and L.L. collected *B. napus* cultivars. J.W. wrote the manuscript. All authors have read and approved the final manuscript.

## Competing interests

The authors declare no competing interests.
