## [Peer Review File · Nature Communications]

Natural variation in BnaA9.NF-YA7 contributes to drought tolerance in *Brassica napus* L.Reviewers' Comments:

Reviewer #1:

Remarks to the Author:

The manuscript by Wang et al. investigates a functional role for a poorly characterized member of the NFYA family in drought stress resistance in Brassica; this is suggested by snps occurring in the promoter and coding sequence and by association with high moisture holding capacity trait. Reduced expression, and, supposedly, reduced TA potential of selected variants might explain higher resistance -by impacting at different levels on YA7 expression itself- an ABF induced negative feedback loop, control of stomatal opening/closure and transpiration, together with regulation of histone H3K36me3 modifications of different target genes negatively impacting on their expression.

Several issues arise from the presented data involving both the approaches used and selection of experimental controls.

Major

Being the shorter isoform of NFYA homologs, which lacks a large portion of the N-terminal (TA region) transactivation potential of YA7 in its allelic variants needs to be analysed in parallel with functional YA homologs previously characterised, to verify its eventual repressive activity on validated target promoters.

H3K36me3, which is evidently globally misregulated in KO lines (as shown in fig 7A), should be analysed on identified target genes. This should be also performed on different Haps (i.e. Hap3/4 - expressing YAc/YAg protein variants) in DS conditions, to validate its possible misregulation in drought tolerant variants. In fact, the limited variation of this modification in YA7 OE plants is not consistent with the dramatic phenotypic consequences observed as opposed to KOs, questioning its inferred relevance for YA7-mediated regulatory mechanisms of drought tolerance.

The OE strategy to dissect NF-Y subunit physiological functions is questionable, given the large number of HFD trimeric complexes with similar (or even different) DNA binding specificities and the variety of processes in which they are involved.

The ChIP-seq and RNAseq data cannot be evaluated as depicted: often, Abs used/ChIP target are not indicated, tracks scales (TPMs) and distances of genomic regions are absent. Tracks with no signal are presented: these data are simply unintelligible.

More specifically.

Abstract:

As endogenous translational/protein levels are not evaluated, this aspect should not be mentioned

Fig 1- related results: Characterization of NF-YA7 Haplotypes

- i. whether variants belonging to each Hap have similar general morphology/phenotype
- ii. Is there any compensation in gene expression? Which are the relative expression levels of alleles A09 and C05 in each Hap/accession and whether basal expression levels of A09 in the different haps are comparable. (sup figs 1a and 4a??)
- iii. A09-YA7 snp S9 is associated with high MHC and drought resistance, it is in strong LD with promoter snp (reduced levels of A9 expression)
Is the promoter (CCAAT) snp alone associated with high MHC? - not clear which is the genotype of each hap -

Fig 2. Transgenic plants generation

i. Which is the genetic background (consequent haplotype) of the WT plant? is A09(C05)-YA7 expressed and is it relevant to DR/KO its expression?

-no apparent phenotypic differences in KO#9 (AAcc), as compared to KO#7,2 aaCC, aacc: why to conclude A (A09) alleles are important in DS tolerance.

ii. Are the siRNA Tg plants analysed? not clear in text/figures.

iii. OE strategy is questionable for defining specific roles of (any) YA subunit.

Transactivation or repression of genes might involve direct competition with other NF-Y and/or CCT/HFD-based trimeric complexes.

The promoter driving YA7 expression in Tg OE lines should be clearly stated (suppl Fig 6).

It is further critical to prove TA capabilities of YA7, as compared to other YA homologs.

Similar phenotypes observed for OE YA7g (Hap3 variant) and YA7c (Hap4 variant) are in line with repressive role of YA7 and/or potential competition with other activatory (NF-Y) complexes; nonetheless, KO lines #2,7 (as compared to #9 when included) do not show critical specific roles in repression for physiological traits analysed.

iv. Statistically relevant differences are absent in WT and tg plants following ABA treatments, and in untreated KO lines leaves (Figs 2j-m).

AAcc phenotype (KO line #9) should also be included in analyses shown in figs h-j to conclude that the A09-YA7 alleles are indeed responsible for the observed stomatal and stromal conductance phenotype in KO lines.

Fig 3. NF-Y regulation of the NF-YA7 promoter allelic variants (Hap3/4) demonstrates it is dependent on CCAAT integrity/promoter snp.

i. Transactivation potential of YA7 should be evaluated in parallel with YA5 and YA9.

Conversely, YA5 and 9 activity should be compared to that of YA7g (Hap3 variant) and YA7c (Hap4 variant) in fig 6d reporter assays on DOR promoter (see comments fig 6)

Fig 4. Reciprocal/negative feedback regulatory mechanisms of NF-YA7 and ABFs

i. It is not clear why ABFs regulation of YA7 -if relevant in establishing drought tolerance- would be similar on Hap3/4 promoters (independently on CCAAT presence) in the establishment of different negative feedback loop in the two Haps; TA potential of C/G alleles should be tested on ABF3 promoter in Luc assays;

YA7 expression should be analysed by RT-qPCR in ABF3 OE lines (sup figure 12)

ii. ChIP-qPCR analysis YA7 promoter scheme should include bp distances from TSS, in order to be compared with deletional characterization in GUS reporter expression analysis of fig 3.

Fig 5.

i. ChIP-seq data: is YA7 overexpressed? if so, is it relevant to induce DS?

no significant differences in ABF3 DNA binding/expression levels.

different tracks refer to? track 4 seems RNA-seq experiment.

ii. DNA binding specificity of NF-Y complexes on ABF promoter CCAAT sites is not determined in shown EMSA, as mutant probes equally compete for binding the target DNA.

Fig 6.

i. see comment Fig 3. reporter assay on DOR promoter should include functional NF-YA homologs: YA5/9 activity should be compared to that of YA7g/c (Hap3/4 variants)

ii. DNA binding specificity of NF-YA7g complexes on DOR promoter CCAAT sites is not determined in shown EMSA, as mutant probes equally compete for binding the target DNA.

iii. Is DOR expressed in OE lines only? please define scale of tracks in RNAseq. (no signal??)

Fig. 7. Global deregulation of H3K36me3 (upregulation in KO lines, unchanged/slightly reduced levels

in OE lines) does not seem to provide a sufficient proof of specific regulatory roles of YA7 on target genes, as there is no provided correlation with expression levels of different targets. H3K36me3 status should be established for all NF-YA7 targets analysed -as done for ABFs- such as NF-YA7 itself, DOR and it should include non-relevant genomic loci.

Reviewer #2:

Remarks to the Author:

Review of NCOMMS-23-39421 "Natural variation in BnaA9.NF-YA7 contributes to drought tolerance in Brassica napus L." by Dr Liu and colleagues.

The authors use accessions from their previous publication of resequencing in a B.napus diversity panel (Ref 18 Lu et al 2019), to identify "48 variation sites covering the BnaA9.NF-YA7 promoter (2 kb) and gene" based on their previous work (ref 18). The authors further mine data from their previous GWA study (ref 17, Tan et al. Euphytica 2017 213:40) and conclude that "BnaA9.NF-YA7 is upregulated by drought stress". This then is followed by a complex series of experiments to find out how this gene is regulated and what role it might have in drought tolerance.

Overall, the manuscript provides interesting insights into a gene pathway that controls stomatal closure and water status under drought stress. But it does not solve the "drought stress tolerance problem" because the focus is on one mechanism of drought tolerance only, that is, stomatal closure under drought stress. As the authors point out, this is a finely balanced process with many checks and balances in the gene pathway.

The main criticism of the paper I have is in the Abstract and Discussion, where the authors justify their research based on the need to genetically improve drought tolerance in B. napus. Line 27 (Abstract) "Taken together, the results show that these alleles that were abandoned during B. napus domestication can serve as direct targets for both genetic engineering and selection for B. napus drought tolerance improvement." Similar comments are made in Discussion. Interestingly, in Discussion, the authors also point out a very important side-effect of genes which close stomata in response to drought stress – and this is "...overexpression of drought-responsive genes often results in growth deficits and yield loss."

There are many reasons why the "C allele" described in this paper is relatively rare in the genetic diversity panel; the authors point out one reason in Discussion - because rapid closure of stomata may be associated with growth deficits and yield loss. It is definitely not justified to suggest that the C allele was abandoned during domestication (early farmers were selecting consciously or unconsciously for consistency of yield in their environment, and if the C allele had a negative impact on yield or yield stability then it could well have been selected against over time). Other reasons for the low frequency of the C allele in the diversity panel might be that the population suffered from genetic drift with random loss of the C allele, or the C allele could be relatively recent mutation, or the genetic diversity panel is biased and not representative of the species. This paper should focus on the fine tuning of the balance between opening and closing stomata under drought stress, and there is evidence in this study that the genes in the pathway are closely balanced to manage drought stress. Most readers with experience in drought stress will doubt that any single genetic alteration will solve the drought tolerance "problem" – and the authors should acknowledge this in Abstract and Discussion.

Line 75 and throughout the manuscript, it is not clear what the authors mean by "moisture holding capacity (MHC)". MHC is normally measured on soil, but clearly it is meant to indicate something about plant tissue here, but the authors do not define here what they mean, nor do the papers they cite in Methods (refs 8, 37) describe MHC. In Fig 1a, what was measured to generate MHC and QTL data?

Similarly, Line 85 refers to "physiologic index analysis showed significant differences among different haplotypes haplotypes, especially Hap3 and Hap4 (Fig. 1c, d)" – what does "physiologic index analysis" mean? It is not described in the Methods.

There is a complex series of experiments which attempt to determine the upstream and downstream functions of genes in the genetic diversity panel. I have tried to follow the logic here:

Line 162: the authors conclude "These results indicated that BnaA9.NF-YA7 negatively regulates drought tolerance by increasing stomatal conductance and transpiration rate."

Line 225: The authors conclude "Together, these data clearly indicate that BnaA9.NF-YA7 functions downstream of BnaABF3/4s."

Line 288: The results suggested "that BnaA9.NF-YA7 represses the BnaABF3/4-regulated pathway through trimerization in response to drought stress."

Line 298: "In comparison with the WT, overexpression of BnaC7.ABF3 alone significantly increased survival because of increased leaf moisture holding capacity and decreased transpiration rate and water loss rate" and "These results suggest that there may be a negative feedback regulatory loop between BnaA9.NF-YA7 and BnaC7.ABF3."

Line 309: "BnaA4.DOR, which encodes the F-box protein DOR that functions as an inhibitory factor for ABA-induced stomatal closure under drought stress^{21,22}, was significantly upregulated in the BnaA9.NF-YA7-OE lines but scarcely in the other genotypes."

Line 329: "These results may explain why overexpression of BnaA9.NF-YA7 increases ABA sensitivity but does not significantly induce stomatal closure."

Line 347: "BnaA9.NF-YA7 regulates the expression of BnaABF3s by controlling H3K36me3 levels."

DISCUSSION:

The authors need to repeat the significance of the G to C mutation at S9_19159584 (M63I) as described in the Results. This mutation from G to C was associated with higher MHC after drought treatment (line 88) in the B napus diversity panel, and this must be spelled out again here.

Line 364: "These results indicate that the co-existence of the M63I substitution and CCAAT-box deletion is the fundamental reason for the strong drought tolerance of Hap4." This is an association but it is not proof of function.

Line 395 points out the dilemma faced by breeders for drought tolerance – should they select for rapid closure of stomata, as appears to be the case in this manuscript? Because, "...overexpression of drought-responsive genes often results in growth deficits and yield loss." This brings into question the evolutionary significance of the genes evaluated in this paper – they may help the plant survive drought by closing stomata, but this also reduces growth and yield.

Line 443: The authors highlight a proposed mechanism: "Low BnaA9.NF-YA7 expression in Hap4... result in higher BnaABF3/4s levels and a lower BnaA4.DOR level, consequently decreasing the transpiration rate and water loss rate, increasing the leaves' moisture holding capacity and leading to stronger drought tolerance."

Line 444 (and in Abstract): the authors are not correct to state that "the M63I substitution and CCAAT-box deletion that increase drought tolerance were abandoned during B. napus domestication" – the reason for their relatively low frequency in this study could be simple genetic drift in isolated populations, or bias in the sample of genotypes, or potentially there are other negative effects of these alleles – such as growth reduction and yield loss. These other plausible reasons should be highlighted!

Reviewer #3:

Remarks to the Author:

In this study, Wang et al. identified several natural variations in Bna.NF-YA7, encoding a nuclear factor Y, which appear to be associated with drought tolerance in B. napus. Specifically, a two base polymorphism in its promoter, within a CCAAT cis-element, alters the expression level of Bna.NF-YA7. They found that BnaA9.NF-YA7 negatively regulates B. napus drought tolerance by promoting BnaA4.DOR expression and in turn restricting ABA-induced stomatal closure. Further analysis by the authors showed that BnaABF3/4s, ABA-responsive element (ABRE)-binding factors, directly promote the transcription of BnaA9.NF-YA7, whereas the latter inhibits the expression of BnaABF3/4s by directly targeting BnaABF3/4s and reducing the H3K36me3 level. Interestingly, Hap4 of BnaA9.NF-YA7 was found to be lost during domestication. Overall, the work provides some new understanding of the molecular mechanisms of drought tolerance of B. napus, providing a potential molecular target to breed drought-tolerant B. napus lines. However, several major concerns must be addressed to

strengthen the manuscript's conclusions and provide a clearer understanding of the reported findings.

Major points:

1. Since BnaA9.NF-YA7 is a gene with natural variations found from GWAS and haplotype analysis, the authors should clearly describe the haplotype of ZS 11, the wild type plant used in this study. According to the methods (L511), the promoters of Hap3 and Hap4 were both cloned from ZS 11. Is ZS 11 heterozygous of Hap3 and Hap4? In Fig. 2a, WT is drought tolerant, similar to Hap1 or Hap4 in Fig. 1c. If ZS 11 is Hap4, it is likely that its low expression level of BnaA9.NF-YA7 and strong drought tolerance are the reasons why the phenotypes of knocked-out lines (aaCC, AAcc, and aacc) in Fig. 2 are not significant compared with WT, not simply due to genetic redundancy as the authors speculated. In this situation, selecting the Hap3 variety for gene knockout and phenotypic comparison is recommended to verify the function of BnaA9.NF-YA7. However, if ZS 11 is Hap3, the authors need to explain why the phenotypes are different between Fig. 1c and Fig. 2a. Is it due to inconsistent treatment conditions, e.g. different durations of drought stress? The treatment conditions should be clarified.

2. The authors claimed that the co-existence of the M63I substitution and CCAAT-box deletion in BnaA9.NF-YA7 is the cause of the strong drought tolerance of Hap4. However, this study mainly showed the effect of the CCAAT-box deletion on downregulating transcription of BnaA9.NF-YA7, while the influence of M63I was not convincingly demonstrated, because of the phenotypes of BnaA9.NF-YA7G and BnaA9.NF-YA7C OX lines in Fig. 2f are similar. Besides the difference in transcriptional activation of pBnaA4.DOR detected in Fig. 6d, the authors should provide more experimental evidence to demonstrate the assumed important role of the M63I substitution.

3. The authors stated that BnaA9.NF-YA7 has a feedback regulation on BnaABF3/4 by inhibiting the transcription of BnaABF3/4 through physically binding the BnaABF3/4 promoter as well as reducing the H3K36me3 levels. However, the following questions remain:

- 1) BnaA9.NF-YA7 directly binds the promoter of BnaABF3/4, does it directly inhibit the expression of ABF3/4? Transient dual-luciferase assays should be performed, like Fig. 4h and Fig. 6d.
- 2) RNA-seq and ChIP-seq showed that BnaA9.NF-YA7 directly binds the promoter of Bna.ASHH4 and represses its expression, and meanwhile, the H3K36me3 level of BnaABF3 was decreased. However, whether the decreased expression of Bna.ASHH4 is the direct cause of the H3K36me3 level change of BnaABF3? More evidence should be provided for this conclusion.

4. According to Fig. 4 and Supplementary Fig. 18, BnaC7.ABF3 promotes the expression of BnaA9.NF-YA7, while the latter inhibits the expression of the former. When both of them were overexpressed in the DO transgenic line in Fig. 5f, how does the expression level of each gene change compared with WT? How to explain that BnaA9.NF-YA7 acts downstream, but the drought tolerance of the DO line is significantly stronger than WT and more similar to BnaC7.ABF3-OX line?

Other comments:

1. In the abstract, 'two bases polymorphism' should be 'two base polymorphism', and 'a CCAAT cis elements' should be 'a CCAAT cis element';
2. L225 'supplementary (Fig. 12c)' should be '(Supplementary Fig. 12c)';
3. Fig. 6j is mislabeled as Fig. 6f;
4. The discussion is largely a repetition of the results and should focus more on the findings' novel aspects.

Reviewer #4:

Remarks to the Author:

In this paper, the authors identified that the natural variation in BnaA9.NF-YA7 were associated with drought tolerance in *B. napa*. They revealed that BnaA9.NF-YA negatively regulated drought tolerance by suppressing the BnaABF genes and their downstream genes. In the drought-resistant traits, mutation of the CCAAT sequence in the BnaA9.NF-YA7 promoter causes low gene expression, and the M63I substitution decreased its transcriptional activity to the target genes. They concluded that these mutations could lead to less function of BnaA9.NF-YA7 and then decrease of the transpiration rate and water loss rate. The large amount of data supports that BnaA9.NF-YA7 negatively regulates desiccation tolerance in *B. napa*. Negative transcriptional feedback with NF-Y factors in the ABA signaling pathway is very interesting. On the other hand, the contribution of the two mutations, especially the M63I mutation to drought resistance, is less clear, because most of the data were analyzed by using BnaA9.NF-YA7-overexpressing or -knockout plants. The authors need to show that this feedback regulation is different among the Hap. I'm not also sure how the M63I mutation affects expression of the BnaA9.NF-YA7-downstream gene in planta. The authors should analyze gene expression and drought tolerance in the BnaA9.NF-YA7 (Hap4) OX. In addition, there are some parts that the readers cannot follow due to lack of clear explanation (or low quality of the experiments) in the Results section and figures.

In Line 66, the authors described "Further mining previously published data showed that ...", but detail of the mining was not explained. Why was BnaA9.NF-YA7 newly found as a good candidate?

There are some cases where the plant appearance in the photo does not match the survival rate in the drought tolerance tests. For example, in Fig. 1c, some leaves are healthy on Hap2, whereas all leaves in Hap3 and Hap5 plants seemed to be shriveled. However, in Fig1d, Hap2 and Hap5 have the same survival rate, and Hap3 has a lower survival rate. Why do these differences occur? Please clarify the criteria for determining whether the plants are survived or dead.

In Supplementary Fig 9a, I'm not sure where the fluorescence of YFP and DAPI are localized. Please show more clear images.

There are other NF-Ys than BnaA9-B2 and BnaA6-C9 in the B2 subgroup of Supplementary Fig. 11a. Why did the authors test the combination of BnaA9-B2 and two C9s in Y2H, and those of two NF-YAs in Y3H? I'm also not sure how to identify NF-YB2 and NF-YCs that could form a trimer complex with BnaA9.NF-YA7 in Supplementary Fig. 16.

I couldn't follow what Supplementary Fig. 16d meant. Why did addition of Met cause inhibition of yeast growth?

Minor points,

Why do yeast colonies in the left panel of Supplementary Fig.16d look blue?

The one-letter abbreviation of tryptophan is W.

Measurement of the soluble protein and sugar content, and other physiological indicators that were written in the Methods, were not performed in the manuscript.

Method of the FRET analysis was not found in the manuscript.

REVIEWER COMMENTS

Reviewer #1 (Remarks to the Author):

The manuscript by Wang et al. investigates a functional role for a poorly characterized member of the NFYA family in drought stress resistance in Brassica; this is suggested by snps occurring in the promoter and coding sequence and by association with high moisture holding capacity trait. Reduced expression, and, supposedly, reduced TA potential of selected variants might explain higher resistance -by impacting at different levels on YA7 expression itself- an ABF induced negative feedback loop, control of stomatal opening/closure and transpiration, together with regulation of histone H3K36me3 modifications of different target genes negatively impacting on their expression.

Several issues arise from the presented data involving both the approaches used and selection of experimental controls.

Response: Thank you for your input.

Major

Being the shorter isoform of NFYA homologs, which lacks a large portion of the N-terminal (TA region) transactivation potential of YA7 in its allelic variants needs to be analysed in parallel with functional YA homologs previously characterised, to verify its eventual repressive activity on validated target promoters.

Response: Thank you for your advice. We obtained fragments of different lengths of *BnaA9.NF-YA7*, including the fragment which lacks a large portion of the N-terminal (TA region). The Y2H (self-activation) assay and dual-luciferase reporter assay showed that *BnaA9.NF-YA7* was an activating transcription factor and the M63I substitution occurred in a Pro-rich transcriptional activation domain, resulting *BnaA9.NF-YA7* had different transcriptional activation abilities in Hap3 and Hap4. We noticed that fragments lacking a large portion of the N-terminal (TA region) do not have transcriptional activation/inhibition ability, therefore parallel analysis was not conducted. The results have been added to the revised manuscript, and the images are shown as follows or can be found in Supplementary Fig. 3d-h of the revised manuscript. The corresponding content is as follows:

“Several *BnaA9.NF-YA7* fragments were designed based on the domain structure of *BnaA9.NF-YA7* (Supplementary Fig. 3d). The Y2H (self-activation) assay and the dual-luciferase reporter assay showed that *BnaA9.NF-YA7* had different transcriptional activation abilities in Hap3 and Hap4 (Supplementary Fig. 3e-h), and the M63I substitution occurred in a Pro-rich transcriptional activation domain (Supplementary Fig. 3a).”

H3K36me3, which is evidently globally misregulated in KO lines (as shown in fig 7A), should be analysed on identified target genes. This should be also performed on different Haps (i.e. Hap3/4 - expressing YAc/YAg protein variants) in DS conditions, to validate its possible misregulation in drought tolerant variants. In fact, the limited variation of this modification in YA7 OE plants is not consistent with the dramatic phenotypic consequences observed as opposed to KOs, questioning its inferred relevance for YA7-mediated regulatory mechanisms of drought tolerance.

Response: Thank you for your advice. We have added the H3K36me3 status detection of *BnaA7.DOR* and *BnaA2.ACT7*, as shown in the response to the comment on Fig7. For ‘the limited variation of this modification in YA7 OE plants is not consistent with the dramatic phenotypic consequences observed as opposed to KOs’, this may be a misinterpretation resulting from visual error. After carefully scrutinizing the quantification results of protein levels in Fig. 6b, although statistical analysis for differences cannot be conducted due to data peculiarities, it is still evident from the histogram that H3K36me3 is upregulated in KO lines and significantly reduced in OE lines. Hence, there exists a significant inverse correlation between the state of H3K36me3 and the expression level of *BnaA9.NF-YA7*.

The OE strategy to dissect NF-Y subunit physiological functions is questionable, given the large number of HFD trimeric complexes with similar (or even different) DNA binding specificities and the variety of processes in which they are involved.

Response: Thank you for your comment. As you mentioned, due to the unique nature of NF-Y trimeric complexes, there are a large number of HFD trimeric complexes with similar (or even different) DNA binding specificities in plants. Consequently, there may be direct competition from other NF-Y and/or CCT/HFD-based trimers for the activation or inhibition of downstream genes. It is questionable whether employing a singular overexpression (OE) strategy would dissect the physiological function of NF-Y subunits. However, in this study, we have employed a combination of OE and KO strategies which aligns with our expectations regarding drought tolerance phenotypes. On the other side, when studying other functional genes, overexpressing target genes may also exert an influence on interactions or competitions. Furthermore, there is an abundance of research utilizing the OE strategy for investigating NF-Y subunits:

- “1. Yang et al. Transcription factors ZmNF-YA1 and ZmNF-YB16 regulate plant growth and drought tolerance in maize. *Plant Physiol.* 2022, 190(2):1506-1525.
2. Mei et al. Maize transcription factor ZmNF-YC13 regulates plant architecture. *J Exp Bot.* 2021, 72(13):4757-4772.
3. Lu et al. Nuclear factor Y subunit GmNFYA competes with GmHDA13 for interaction with GmFVE to positively regulate salt tolerance in soybean. *Plant Biotechnol J.* 2021, 19(11):2362-2379.
4. Wang et al. NF-Y plays essential roles in flavonoid biosynthesis by modulating histone modifications in tomato. *New Phytol.* 2021, 229(6):3237-3252.
5. Zhou et al. Root-specific NF-Y family transcription factor, PdNF-YB21, positively regulates root growth and drought resistance by abscisic acid-mediated indoleacetic acid transport in *Populus*. *New Phytol.* 2020, 227(2):407-426.
6. Xiong et al. NF-YC12 is a key multi-functional regulator of accumulation of seed storage substances in rice. *J Exp Bot.* 2019, 70(15):3765-3780.
7. Sato et al. NF-YB2 and NF-YB3 Have Functionally Diverged and Differentially Induce Drought and Heat Stress-Specific Genes. *Plant Physiol.* 2019, 180(3):1677-1690.
8. Bi et al. Overexpression of the transcription factor NF-YC9 confers abscisic acid hypersensitivity in *Arabidopsis*. *Plant Mol Biol.* 2017, 95(4-5):425-439.
9. Liu et al. The NF-YC-RGL2 module integrates GA and ABA signalling to regulate seed germination in *Arabidopsis*. *Nat Commun.* 2016, 14;7:12768.
10. Sato et al. *Arabidopsis* DPB3-1, a DREB2A interactor, specifically enhances heat stress-induced gene expression by forming a heat stress-specific transcriptional complex with NF-Y subunits. *Plant Cell.* 2014, 26(12):4954-73.”

The ChIP-seq and RNAseq data cannot be evaluated as depicted: often, Abs used/ChIP target are not indicated, tracks scales (TPMs) and distances of genomic regions are absent. Tracks with no signal are presented: these data are simply unintelligible.

Response: Thank you for your input. We have added information including Abs, tracks

scales, distances of genomic regions, etc. to the ChIP-seq and RNA-seq data, and the modified results can be found in Fig. 5b, 7a, and Supplementary Fig. 16a.

More specifically.

Abstract:

As endogenous translational/protein levels are not evaluated, this aspect should not be mentioned

Response: Thank you for your valuable advice. We have deleted the relevant content in the revised manuscript.

Fig 1- related results: Characterization of NF-YA7 Haplotypes

i. whether variants belonging to each Hap have similar general morphology/phenotype

Response: The limited contribution of a single gene to the drought-resistant phenotype is widely acknowledged. Consequently, even within drought-sensitive haplotypes, there exist a few drought-resistant materials, resulting in each haplotype's material does not have a completely similar phenotype.

ii. Is there any compensation in gene expression? Which are the relative expression levels of alleles A09 and C05 in each Hap/accession and whether basal expression levels of A09 in the different haps are comparable. (sup figs 1a and 4a??)

Response: In our previous study, we analyzed the expression levels of *BnaA9.NF-YA7* and *BnaC5.NF-YA7* in each haplotype. As depicted in the figure below, it is evident that the basal expression level of *BnaA9.NF-YA7* in Hap4 exhibits a significant decrease compared to other haplotypes. Conversely, there are no notable differences observed in the expression level of *BnaC5.NF-YA7* across all haplotypes. Despite their high homology and potential functional redundancy, we have confirmed the absence of compensatory expression between *BnaA9.NF-YA7* and *BnaC5.NF-YA7*.

iii. A09-YA7 snp S9 is associated with high MHC and drought resistance, it is in strong LD with promoter snp (reduced levels of A9 expression)

Is the promoter (CCAAT) snp alone associated with high MHC? - not clear which is the genotype of each hap -

Response: Our unclear description may have caused confusion for you. In this study, *BnaA9.NF-YA7^C* alone associated with high WHC (MHC revised to WHC), the

accessions carrying the C allele were further divided into two types (Hap1 and Hap4), and deletion of the CCAAT-box, reducing the *BnaA9.NF-YA7* expression level, occurred only in Hap4. A clearer description can be found in the revised **Discussion** section, **lines 393-396**.

Fig 2. Transgenic plants generation

i. Which is the genetic background (consequent haplotype) of the WT plant? is A09(C05)-YA7 expressed and is it relevant to DR/KO its expression?

-no apparent phenotypic differences in KO#9 (AAcc), as compared to KO#7,2 aaCC, aacc: why to conclude A (A09) alleles are important in DS tolerance.

Response: In this study, we used Zhongshuang 11 (ZS11) as the WT, which belongs to Hap3. In ZS11 background, higher levels of *BnaA9.NF-YA7* transcripts were detected in embryos, leaves, and seeds compared with other tissues, similar to the result for *BnaC5.NF-YA7* (Supplementary Fig. 5a). *BnaA9.NF-YA7* expression was significantly induced by drought stress, and the expression level of Hap3 (including ZS11) was significantly higher than that of Hap4. The expression of *BnaC5.NF-YA7* was also induced by drought, but the level of *BnaC5.NF-YA7* was not significantly different among the five haplotypes. For KO lines, we did not detect the expression levels of *BnaA9.NF-YA7* and *BnaC5.NF-YA7*, as the mutate site happens in the coding region.

Throughout the manuscript, we emphasize that *BnaA9.NF-YA7* and *BnaC5.NF-YA7* are functionally redundant, including in response to drought. This may also be the reason why there is no phenotypic difference between KO#2, KO#7 and KO#9. The selection of *BnaA9.NF-YA7* as the target gene in this study was not based on the importance of *BnaA9.NF-YA7* over *BnaC5.NF-YA7* in drought tolerance, but rather on the detection of relevant variations specifically in *BnaA9.NF-YA7* that were found to be associated with drought tolerance. Conversely, these variation sites were not identified in *BnaC5.NF-YA7*.

ii. Are the siRNA Tg plants analysed? not clear in text/figures.

Response: In this study, only phenotypic observations were conducted on all RNAi plants, and the results are shown in Supplementary Fig. 8c-e (Supplementary Fig. 7 has been changed to Supplementary Fig. 8 in revised manuscript).

iii. OE strategy is questionable for defining specific roles of (any) YA subunit.

Transactivation or repression of genes might involve direct competition with other NF-Y and/or CCT/HFD-based trimeric complexes.

The promoter driving YA7 expression in Tg OE lines should be clearly stated (suppl Fig 6).

It is further critical to prove TA capabilities of YA7, as compared to other YA homologs. Similar phenotypes observed for OE YA7g (Hap3 variant) and YA7c (Hap4 variant) are in line with repressive role of YA7 and/or potential competition with other activatory (NF-Y) complexes; nonetheless, KO lines #2,7 (as compared to #9 when included) do not show critical specific roles in repression for physiological traits

analysed.

Response: Thank you for your comment. For “OE strategy is questionable for defining specific roles of (any) YA subunit”, we have already responded in the **Major** comments, and here we briefly state again: in this study, we adopt a strategy combining OE and KO to elucidate the potential molecular functions of NF-Y subunits in drought response. Although there are a large number of HFD trimeric complexes with similar (or even different) DNA binding specificities in plants, thus there may be direct competition from other NF-Y and/or CCT/HFD-based trimers for the activation or inhibition of downstream genes. The results showed that OE and KO exhibited completely opposite phenotypes in drought tolerance experiments. Furthermore, in research reports on the NF-Y subunit, a large number of scholars have used the OE strategy.

In the **Methods** section, we have already introduced the detailed information of *BnaA9.NF-YA7* overexpression vectors, as shown in **lines 494-497**: “The *BnaA9.NF-YA7* overexpression construct was generated with pENTR™ Directional TOPO® Cloning Kits (Invitrogen, USA) according to the manufacturer’s protocols, and the relevant fragment was cloned and inserted into the binary vector pEarleyGate 101.”. Here, we have added the promoter information of *BnaA9.NF-YA7* overexpression to the caption in Supplementary Fig. 7 (Supplementary Fig. 6 has been changed to Supplementary Fig. 7 in revised manuscript).

To prove TA capabilities of *BnaA9.NF-YA7*, we obtained fragments of different lengths of *BnaA9.NF-YA7*, including the fragment which lacks a large portion of the N-terminal (TA region). The Y2H (self-activation) assay and the dual-luciferase reporter assay showed that *BnaA9.NF-YA7* was an activating transcription factor and the M63I substitution occurred in a Pro-rich transcriptional activation domain, resulting *BnaA9.NF-YA7* had different transcriptional activation abilities in *BnaA9.NF-YA7^C* and *BnaA9.NF-YA7^G* (Supplementary Fig. 3d-h).

iv. Statistically relevant differences are absent in WT and tg plants following ABA treatments, and in untreated KO lines leaves (Figs 2j-m).

AAcc phenotype (KO line #9) should also be included in analyses shown in figs h-j to conclude that the A09-YA7 alleles are indeed responsible for the observed stomatal and stromal conductance phenotype in KO lines.

Response: Thank you for your comment. We have also observed this phenomenon, and we hypothesize that the insignificant differences in stomatal conductance and transpiration rate between wild-type and OE plants following ABA treatment as one plant of OE-7# and OE-8# give unusual low value. To ensure data integrity and objectivity, we kept these outlier data. For untreated KO lines leaves, there were no significant differences in stomatal density and stomatal index between the WT and KO lines, thus resulting in no significant differences in stomatal conductance and transpiration rate. The expression level of *BnaA9.NF-YA7* was found to be lower in Hap4 than Hap3, while the M63I mutation led to variations in the transcription activation of *Bna.DORs* between Hap4 and Hap3. The combined effect of these factors resulted in higher drought tolerance for Hap4 compared to Hap3. Under normal conditions, the expression of *Bna.DORs* in KO and WT is very low, without significant difference in between them.

Considering the functional redundancy between *BnaC5.NF-YA7* and *BnaA9.NF-YA7*, we did not check of stomatal related indexes of KO #9 (AAcc) in the previous experiment. The previous data were collected in the field, and due to seasonal factors, we could only culture the experimental materials in a laboratory incubator to supplement the relevant data. However, the great difference in environment between the incubator and the field led to a great difference in results. Here, only the result data is provided for reference.

Fig 3. NF-Y regulation of the NF-YA7 promoter allelic variants (Hap3/4) demonstrates it is dependent on CCAAT integrity/promoter snp.

i. Transactivation potential of YA7 should be evaluated in parallel with YA5 and YA9. Conversely, YA5 and 9 activity should be compared to that of YA7g (Hap3 variant)

and YA7c (Hap4 variant) in fig 6d reporter assays on DOR promoter (see comments fig 6)

Response: Transactivation potential of *BnaA9.NF-YA7* (*BnaA9.NF-YA7^G* and *BnaA9.NF-YA7^C*) had been evaluated in parallel with *BnaA6.NF-YA5* and *BnaC5.NF-YA9*. The finished result is illustrated in Fig 3g. About the result of DOR promoter has been illustrated in Fig 7d (Fig. 6 has been changed to Fig. 7 in revised manuscript). The revised place are as follows: **Results** section, **lines 201-204**.

Fig 4. Reciprocal/negative feedback regulatory mechanisms of NF-YA7 and ABFs

i. It is not clear why ABFs regulation of YA7 -if relevant in establishing drought tolerance- would be similar on Hap3/4 promoters (independently on CCAAT presence) in the establishment of different negative feedback loop in the two Haps; TA potential of C/G alleles should be tested on ABF3 promoter in Luc assays;

Response: *BnaA9.NF-YA7* is upregulated by drought stress and ABA induction, indicating that *BnaA9.NF-YA7* is involved in drought stress response and ABA signaling. In a yeast One-Hybrid screening library based on the promoter fragment of *BnaA9.NF-YA7*, we repeatedly detected the key transcription factor ABFs of ABA signaling. Combined with Y1H, EMSA, CHIP-qPCR, and the dual-luciferase reporter assay, we confirmed that *BnaA9.NF-YA7* is a direct target gene of ABFs, and drought and ABA-induced upregulation of *BnaA9.NF-YA7* is dependent on the regulation of ABFs. The results of Supplementary Fig. 13c (Supplementary Fig. 12 has been changed to Supplementary Fig. 13 in revised manuscript) further confirm our hypothesis that the expression level of *BnaA9.NF-YA7* is downregulated in ABF3-RNAi plants and significantly increased in ABF3 overexpressing plants. Meanwhile, ABA-induced upregulation of *BnaA9.NF-YA7* was significantly inhibited in ABF3-RNAi compared with WT. Based on the above results, we confirmed that ABFs may regulate *BnaA9.NF-YA7*. However, the regulation of ABFs on *BnaA9.NF-YA7* was not different in H3/H4, so in the negative feedback loop, ABF as the upstream of YA7 was consistent in the two haplotypes. One of the reasons for the difference in drought tolerance is the feedback regulation of ABFs by different haplotypes of *BnaA9.NF-YA7*.

For “TA potential of C/G alleles should be tested on ABF3 promoter in Luc assays”, the results had been added to Supplementary Fig. 18e (Supplementary Fig. 17 has been changed to Supplementary Fig. 18 in revised manuscript). The revised place are as follows: **Results** section, **lines 295-299**.

YA7 expression should be analysed by RT-qPCR in ABF3 OE lines (sup figure 12)

Response: The expression of *BnaA9.NF-YA7* in *BnaC7.ABF3* OE lines can be found in Supplementary Fig. 13c (Supplementary Fig. 12 has been changed to Supplementary Fig. 13 in revised manuscript).

ii. ChIP-qPCR analysis YA7 promoter scheme should include bp distances from TSS, in order to be compared with deletional characterization in GUS reporter expression analysis of fig 3.

Response: Thank you for your valuable advice. The bp distances from TSS has been added to the *BnaA9.NF-YA7* promoter scheme in Fig 4.

Fig 5.

i. ChIP-seq data: is YA7 overexpressed? if so, is it relevant to induce DS?

no significant differences in ABF3 DNA binding/expression levels.

different tracks refer to? track 4 seems RNA-seq experiment.

Response: In the **Methods** section, we describe that ChIP-seq uses overexpressed transgenic lines. *BnaA9.NF-YA7* is upregulated by drought stress, but its expression level does not increase under drought stress in overexpressed plants. To prevent any potential misinterpretation, we conducted ChIP-seq analysis using both normal and drought-treated materials.

Among the three homologous genes of ABF3, *BnaC1.ABF3* and *BnaAn.ABF3* are completely inhibited by *BnaA9.NF-YA7* (even under drought stress) (Supplementary Fig. 16a), while *BnaC7.ABF3* is partially inhibited (Fig. 5b). Previously, the ChIP-seq data lacks track scales, making it difficult to comprehend. Now, we have included relevant tracks information in Fig. 5b and Supplementary Fig. 16a. After careful examination of the data again, we are sure that track 4 represents sequencing data from drought-treated material after immunoprecipitation, with the original data have been deposited in the NCBI Sequence Read Archive under BioProject accession numbers PRJNA978979. Additionally, we provide a broader view here.

A larger view of tracks 3 and 4:

ii. DNA binding specificity of NF-Y complexes on ABF promoter CCAAT sites is not determined in shown EMSA, as mutant probes equally compete for binding the target DNA.

Response: We have carefully checked the experimental records and EMSA results, and we confirm that the mutant probes cannot equally compete for binding the target DNA.

In EMSA assay, we only added the mutant probes in lanes 8 and 12 (The bottom image is lane 11) (the above lanes did not add the target probes). Electrophoresis analysis revealed the absence of any discernible shift band in lanes 8 and 12 (The bottom image is lane 11) (Fig 5e), thereby substantiating that the mutant probes cannot equally compete for binding the target DNA.

Fig 6.

i. see comment Fig 3. reporter assay on DOR promoter should include functional NF-YA homologs: YA5/9 activity should be compared to that of YA7^{G/C} (Hap3/4 variants)
Response: Thanks for your advice. The result of BnaA6.NF-YA5 and BnaC5.NF-YA9 activity compared to that of BnaA9.NF-YA7^G and BnaA9.NF-YA7^C in reporter assays on DOR promoter has been added to Fig 7d (Fig. 6 has been changed to Fig. 7 in revised manuscript) in revision.

ii. DNA binding specificity of NF-YA7g complexes on DOR promoter CCAAT sites is not determined in shown EMSA, as mutant probes equally compete for binding the target DNA.

Response: We have carefully checked the experimental records and EMSA results, and we confirm that the mutant probes cannot equally compete for binding the target DNA. Unlike Fig. 5e, we added both a mutation probe and a target probe in lane 7. Compared to the shift band in lane 6, there was no significant change in the shift band in lane 7. Meanwhile, only mutant probes were added to lane 5 without target probes, and no shift bands were detected in the electrophoresis results (Fig. 7b, Fig. 6 has been changed to Fig. 7 in revised manuscript). These results indicate that the mutant probes cannot equally compete for binding the target DNA.

iii. Is DOR expressed in OE lines only? please define scale of tracks in RNAseq. (no signal??)

Response: *BnaA4.DOR* is only expressed in *BnaA9.NF-YA7* overexpressing lines, and no expression was detected in other lines, so there is no signal in tracks of RNA-seq. We have redefined the scale of tracks in Fig 7a (Fig. 6 has been changed to Fig. 7 in revised manuscript).

Fig. 7. Global deregulation of H3K36me3 (upregulation in KO lines, unchanged/slightly reduced levels in OE lines) does not seem to provide a sufficient proof of specific regulatory roles of YA7 on target genes, as there is no provided

correlation with expression levels of different targets.

H3K36me3 status should be established for all NF-YA7 targets analysed -as done for ABFs- such as NF-YA7 itself, DOR and it should include non-relevant genomic loci.

Response: Thanks for your valuable advice. We have supplemented the H3K36me3 status of *BnaA7.DOR* and added the H3K36me3 status of *BnaA2.ACT7*. As shown in the figure below, there was no significant difference in H3K36me3 status between *BnaA7.DOR* and *BnaA2.ACT7* in *BnaA9.NF-YA7* overexpressing and knockout plants. The relevant results can be found in Supplementary Fig. 21a.

The revised place is as follows: **Results** section, **lines 331-344:**

“Comparable changes were not observed in chromatin at the *BnaA2.ACT7* locus, which was used as a negative control (Supplementary Fig. 21a). Next, we performed virus-induced *Bna.ASHH4s* silencing using a tobacco rattle virus (TRV)-based system to reduce *Bna.ASHH4s* mRNA levels and determined the expression level of *BnaC1.ABF3* and *BnaC7.ABF3* in TRV2-*Bna.ASHH4* plants. The results indicated that *BnaC1.ABF3* and *BnaC7.ABF3* expression were positively correlated with *Bna.ASHH4s* expression. The expression of *BnaC1.ABF3* and *BnaC7.ABF3* in TRV2-*Bna.ASHH4* plants did not significantly increase even under dehydration conditions (Supplementary Fig. 21b). In turn, H3K36 trimethylation at *BnaC1.ABF3* and *BnaC7.ABF3* showed significant reduction in the TRV2-*Bna.ASHH4* lines (Supplementary Fig. 21c). In addition, the expression level of *BnaA9.NF-YA7* was not affected by the downregulation of *Bna.ASHH4s* under normal condition. Taken together, these results indicate that *BnaA9.NF-YA7* modulates the expression of *BnaABF3s* by controlling H3K36me3 levels.”

Results section, **lines 353-355:**

‘However, ChIP analysis of H3K36 trimethylation at the *BnaA4.DOR* locus in different genotypes showed no significant change in the extent of H3K36 trimethylation at *BnaA4.DOR* (Supplementary Fig. 21a).’

Reviewer #2 (Remarks to the Author):

Review of NCOMMS-23-39421 "Natural variation in *BnaA9.NF-YA7* contributes to drought tolerance in *Brassica napus* L." by Dr Liu and colleagues.

The authors use accessions from their previous publication of resequencing in a *B.napus* diversity panel (Ref 18 Lu et al 2019), to identify “48 variation sites covering the *BnaA9.NF-YA7* promoter (2 kb) and gene” based on their previous work (ref 18). The authors further mine data from their previous GWA study (ref 17, Tan et al. *Euphytica* 2017 213:40) and conclude that “*BnaA9.NF-YA7* is upregulated by drought stress”. This then is followed by a complex series of experiments to find out how this gene is regulated and what role it might have in drought tolerance.

Overall, the manuscript provides interesting insights into a gene pathway that controls stomatal closure and water status under drought stress. But it does not solve the “drought stress tolerance problem” because the focus is on one mechanism of drought tolerance only, that is, stomatal closure under drought stress. As the authors point out, this is a finely balanced process with many checks and balances in the gene pathway.

The main criticism of the paper I have is in the Abstract and Discussion, where the authors justify their research based on the need to genetically improve drought tolerance in *B. napus*. Line 27 (Abstract) “Taken together, the results show that these alleles that were abandoned during *B. napus* domestication can serve as direct targets for both genetic engineering and selection for *B. napus* drought tolerance improvement.” Similar comments are made in Discussion. Interestingly, in Discussion, the authors also point out a very important side-effect of genes which close stomata in response to drought stress – and this is “...overexpression of drought-responsive genes often results in growth deficits and yield loss.”

Response: Thank you for your input. We have corrected this section by changing 'solve drought problems' to 'analysis drought tolerance mechanisms'. The revised place is as follows: **Abstract** section: **lines 22-25:**

“Our findings uncover that *BnaA9.NF-YA7* serves as a supplementary mechanism for ABA signal balance under drought stress conditions in *B. napus*, and provides a potential molecular target to breed drought-tolerant *B. napus* lines.”

Discussion section: **lines 470-473:**

“Although the contribution of *BnaA9.NF-YA7* as a direct target for genetic engineering to improve *B. napus* drought tolerance was finite, our findings provide insights into the mechanisms that control ABA signaling homeostasis, in which *BnaA9.NF-YA7* serves as a supplementary mechanism.”

There are many reasons why the “C allele” described in this paper is relatively rare in the genetic diversity panel; the authors point out one reason in Discussion - because rapid closure of stomata may be associated with growth deficits and yield loss. It is definitely not justified to suggest that the C allele was abandoned during domestication (early farmers were selecting consciously or unconsciously for consistency of yield in their environment, and if the C allele had a negative impact on yield or yield stability then it could well have been selected against over time). Other reasons for the low frequency of the C allele in the diversity panel might be that the population suffered from genetic drift with random loss of the C allele, or the C allele could be relatively recent mutation, or the genetic diversity panel is biased and not representative of the species. This paper should focus on the fine tuning of the balance between opening and

closing stomata under drought stress, and there is evidence in this study that the genes in the pathway are closely balanced to manage drought stress. Most readers with experience in drought stress will doubt that any single genetic alteration will solve the drought tolerance “problem” – and the authors should acknowledge this in Abstract and Discussion.

Response: Thank you for your valuable advice. After re-examining the relevant content, we agree with your viewpoint that the low frequency of the C allele in the diversity panel might be that the population suffered from genetic drift with random loss of the C allele, or the C allele could be relatively recent mutation. Therefore, we have revised the corresponding description in **Abstract** and **Discussion**. At the same time, we also acknowledge that changes in a single gene have limited contribution to solving drought tolerance in rapeseed. The revised region is as follows:

Abstract section: **lines 22-25:**

“Our findings uncover that *BnaA9.NF-YA7* serves as a supplementary mechanism for ABA signal balance under drought stress conditions in *B. napus*, and provides a potential molecular target to breed drought-tolerant *B. napus* lines.”

Discussion section: **lines 425-454, 470-473:**

“These results suggest that the negative feedback regulatory network mediated by *BnaA9.NF-YA7* serves as a supplementary mechanism for ABA signal balance. Stomatal closure preserves water in the plant, and ABA has a role in the regulation of stomatal movement to prevent water loss⁴⁰. Because stomatal movements control CO₂ influx and transpiration, drought resistance strategies to reduce water loss via stomatal closure occur at the expense of photosynthesis, growth, and yield³⁸. Here, we found that overexpression of *BnaA9.NF-YA7* increased ABA sensitivity, but the percentage of closed stomata was not significantly affected by ABA (Fig 21-q). We propose that this strategy may be part of an ABA signaling balance mechanism. *BnaA9.NF-YA7* and its NF-Y complexes directly interact with the *BnaA4.DOR* promoter upregulates its expression level, thereby maintaining partial stomatal opening under drought stress and ensuring the CO₂ influx needed for plant photosynthesis, thus allowing plants to continue to grow under drought stress conditions. The M63I substitution occurred in a Pro-rich transcriptional activation domain, resulting in a transcriptional activation difference between *BnaA9.NF-YA7^G* and *BnaA9.NF-YA7^C* (Fig. 7d), thus breaking the fine tuning of the balance between opening and closing stomata under drought stress. Nevertheless, the impact of the M63I mutation on drought tolerance is limited and only when accompanied by the CCAAT mutation can it further enhance drought tolerance. This may also explain why Hap4 exhibits stronger drought tolerance than Hap1 (Fig. 1c, d). Therefore, we posit that these two mutations are both independent yet intricately linked, with CCAAT primarily influencing the feedback regulation loop, while M63I predominantly modulates stoma sensitivity toward ABA. The excessive expression of *BnaA9.NF-YA7^{G/C}* excessively inhibited the transmission of ABA signals, and although the overexpression of *BnaA9.NF-YA7^C* promotes stomatal closure, it is insufficient to reverse the drought sensitivity caused by the inhibition of ABA signaling (Fig. 2f). Taken together, the data indicate that *BnaA9.NF-YA7* plays a dual role as both an inhibitor and activator, effectively regulating ABA signaling in rapeseed to maintain a

balance between growth and drought stress responses.”

“Although the contribution of *BnaA9.NF-YA7* as a direct target for genetic engineering to improve *B. napus* drought tolerance was finite, our findings provide insights into the mechanisms that control ABA signaling homeostasis, in which *BnaA9.NF-YA7* serves as a supplementary mechanism.”

Line 75 and throughout the manuscript, it is not clear what the authors mean by “moisture holding capacity (MHC)”. MHC is normally measured on soil, but clearly it is meant to indicate something about plant tissue here, but the authors do not define here what they mean, nor do the papers they cite in Methods (refs 8, 37) describe MHC. In Fig 1a, what was measured to generate MHC and QTL data?

Response: Many thanks for your good suggestion. Moisture holding capacity (MHC) is misused here, MHC should be water holding capacity of separated leaves (WHC). A brief description of WHC has been added to the text: “To measure water holding capacity (WHC), fully expanded fourth leaves were detached from the seedlings and the petiole were wrapped with fresh-keeping film to prevent water loss from the petiole. The fresh weight of detached leaves was measured immediately detached from the plants (W_f), 24 h dark treatment ($23 \pm 1^\circ\text{C}$ and 30–40% RH, W_t), and drying (W_d). The water holding capacity of separated leaves was calculated using the formula $\text{WHC} = (W_t - W_d) / (W_f - W_d) \times 100$ and was shown as a percentage. The experiment was repeated three times. For each repeat, at least five leaves from different plants of each genotype were used.”

About Fig 1a, A brief description has been added to the **Methods** section: “A total of 205 overlapping lines from previous reports of Tan *et al.* (2017) and Lu *et al.* (2019) were used for association analysis of WHC^{17,18}. The GWAS was conducted using the mixed linear model (MLM)⁴¹ with a total of 581,532 SNPs (miss data<20%, MAF>0.05) in the GAPIT R package⁴². The Manhattan plot was displayed using the CMplot R package⁴³.”

Similarly, Line 85 refers to “physiologic index analysis showed significant differences among different haplotypes haplotypes, especially Hap3 and Hap4 (Fig. 1c, d)” – what does “physiologic index analysis” mean? It is not described in the Methods.

Response: “physiologic index analysis” refers to the measurement of the relevant physiologic data listed in Supplementary Fig. 2e-h, and the relevant description can be found in the **Methods** section (Original manuscript: lines 496-502, Revision: **lines 537-548**).

There is a complex series of experiments which attempt to determine the upstream and downstream functions of genes in the genetic diversity panel. I have tried to follow the logic here:

Line 162: the authors conclude “These results indicated that *BnaA9.NF-YA7* negatively regulates drought tolerance by increasing stomatal conductance and transpiration rate.”

Line 225: The authors conclude “Together, these data clearly indicate that *BnaA9.NF-YA7* functions downstream of *BnaABF3/4s*.”

Line 288: The results suggested “that *BnaA9.NF-YA7* represses the *BnaABF3/4*-regulated pathway through trimerization in response to drought stress.”

Line 298: “In comparison with the WT, overexpression of *BnaC7.ABF3* alone significantly increased survival because of increased leaf moisture holding capacity and decreased transpiration rate and water loss rate” and “These results suggest that there may be a negative feedback regulatory loop between *BnaA9.NF-YA7* and *BnaC7.ABF3*.”

Line 309: “*BnaA4.DOR*, which encodes the F-box protein DOR that functions as an inhibitory factor for ABA-induced stomatal closure under drought stress^{21,22}, was significantly upregulated in the *BnaA9.NF-YA7*-OE lines but scarcely in the other genotypes.”

Line 329: “These results may explain why overexpression of *BnaA9.NF-YA7* increases ABA sensitivity but does not significantly induce stomatal closure.”

Line 347: “*BnaA9.NF-YA7* regulates the expression of *BnaABF3s* by controlling H3K36me3 levels.”

Response: Thank you for your comment, your mentioned point are our main results.

DISCUSSION:

The authors need to repeat the significance of the G to C mutation at S9_19159584 (M63I) as described in the Results. This mutation from G to C was associated with higher MHC after drought treatment (line 88) in the *B. napus* diversity panel, and this must be spelled out again here.

Response: Thanks for your valuable advice. We emphasized again in the results section that the mutation from G to C is associated with higher WHC (MHC revised to WHC) after drought treatment in the *B. napus* diversity panel, and discussed the significance of the G to C mutation at S9_19159584. The revised regions are as follows:

Discussion section, **lines 393-394.**

“The accessions with the *BnaA9.NF-YA7^C* allele had a higher WHC than the accessions with the *BnaA9.NF-YA7^G* allele under drought treatment.”

Discussion section, **lines 428-442.**

“Stomatal closure preserves water in the plant, and ABA has a role in the regulation of stomatal movement to prevent water loss⁴⁰. Because stomatal movements control CO₂ influx and transpiration, drought resistance strategies to reduce water loss via stomatal closure occur at the expense of photosynthesis, growth, and yield³⁸. Here, we found that overexpression of *BnaA9.NF-YA7* increased ABA sensitivity, but the percentage of closed stomata was not significantly affected by ABA (Fig 21-q). We propose that this strategy may be part of an ABA signaling balance mechanism. *BnaA9.NF-YA7* and its NF-Y complexes directly interact with the *BnaA4.DOR* promoter upregulates its expression level, thereby maintaining partial stomatal opening under drought stress and ensuring the CO₂ influx needed for plant photosynthesis, allowing plants to continue to grow under drought stress conditions. The M63I substitution occurred in a Pro-rich transcriptional activation domain, resulting in a transcriptional activation difference

between *BnaA9.NF-YA7^G* and *BnaA9.NF-YA7^C* (Fig. 7d), thus breaking the fine tuning of the balance between opening and closing stomata under drought stress.”

Line 364: “These results indicate that the co-existence of the M63I substitution and CCAAT-box deletion is the fundamental reason for the strong drought tolerance of Hap4.” This is an association but it is not proof of function.

Response: Thanks for your advice. We have revised the relevant content in the revised manuscript, and the revision results are as follows: **Discussion** section, **lines 396-399**. “However, deletion of the CCAAT-box, which reduced the *BnaA9.NF-YA7* expression level, occurred only in Hap4, suggesting that the M63I substitution cooperates with CCAAT-box deletion and can be used to increase the drought tolerance of *B. napus*.”

Line 395 points out the dilemma faced by breeders for drought tolerance – should they select for rapid closure of stomata, as appears to be the case in this manuscript? Because, “...overexpression of drought-responsive genes often results in growth deficits and yield loss.” This brings into question the evolutionary significance of the genes evaluated in this paper – they may help the plant survive drought by closing stomata, but this also reduces growth and yield.

Response: In this study, the M63I (G to C) substitution is essentially degenerate. Although the majority of *B. napus* varieties have a genotype of G at this locus (556/608), the genotype at this locus is C in their ancestor species and other cruciferous plants. Combining the characteristics of tall rapeseed plants, high water demand, and sensitivity to water deficit, as well as the high *F_{ST}* level in the region where the gene is located, changes in C to G may be a selection strategy that comprehensively considers yield, growth, and water use efficiency.

Line 443: The authors highlight a proposed mechanism: “Low *BnaA9.NF-YA7* expression in Hap4... result in higher *BnaABF3/4s* levels and a lower *BnaA4.DOR* level, consequently decreasing the transpiration rate and water loss rate, increasing the leaves' moisture holding capacity and leading to stronger drought tolerance.”

Response: Thank you for your comment.

Line 444 (and in Abstract): the authors are not correct to state that “the M63I substitution and CCAAT-box deletion that increase drought tolerance were abandoned during *B. napus* domestication” – the reason for their relatively low frequency in this study could be simple genetic drift in isolated populations, or bias in the sample of genotypes, or potentially there are other negative effects of these alleles – such as growth reduction and yield loss. These other plausible reasons should be highlighted!

Response: Thank you for your advice. As you said, there are many reasons for the low frequency of the M63I substitution and CCAAT-box deletion in this study, and without the support of substantial evidence, we have abandoned the discussion of the reasons for the low frequency of the M63I substitution and CCAAT-box deletion, and instead focused on the fine tuning of the balance between opening and closing stomata under drought stress. We have corrected the relevant content in the revised manuscript, and

the detailed revision is as follows: **Abstract** section, lines **11-25** and **Discussion** section, lines **400-454**.

Reviewer #3 (Remarks to the Author):

In this study, Wang et al. identified several natural variations in Bna.NF-YA7, encoding a nuclear factor Y, which appear to be associated with drought tolerance in *B. napus*. Specifically, a two base polymorphism in its promoter, within a CCAAT cis-element, alters the expression level of Bna.NF-YA7. They found that BnaA9.NF-YA7 negatively regulates *B. napus* drought tolerance by promoting BnaA4.DOR expression and in turn restricting ABA-induced stomatal closure. Further analysis by the authors showed that BnaABF3/4s, ABA-responsive element (ABRE)-binding factors, directly promote the transcription of BnaA9.NF-YA7, whereas the latter inhibits the expression of BnaABF3/4s by directly targeting BnaABF3/4s and reducing the H3K36me3 level. Interestingly, Hap4 of BnaA9.NF-YA7 was found to be lost during domestication. Overall, the work provides some new understanding of the molecular mechanisms of drought tolerance of *B. napus*, providing a potential molecular target to breed drought-tolerant *B. napus* lines. However, several major concerns must be addressed to strengthen the manuscript's conclusions and provide a clearer understanding of the reported findings.

Response: Thank you for your input. To strengthen the manuscript's conclusions and provide a clearer understanding of the reported findings, we have addressed most of the concerns in a targeted manner.

Major points:

1. Since BnaA9.NF-YA7 is a gene with natural variations found from GWAS and haplotype analysis, the authors should clearly describe the haplotype of ZS 11, the wild type plant used in this study. According to the methods (L511), the promoters of Hap3 and Hap4 were both cloned from ZS 11. Is ZS 11 heterozygous of Hap3 and Hap4? In Fig. 2a, WT is drought tolerant, similar to Hap1 or Hap4 in Fig. 1c. If ZS 11 is Hap4, it is likely that its low expression level of BnaA9.NF-YA7 and strong drought tolerance are the reasons why the phenotypes of knocked-out lines (aaCC, AAcc, and aacc) in Fig. 2 are not significant compared with WT, not simply due to genetic redundancy as the authors speculated. In this situation, selecting the Hap3 variety for gene knockout and phenotypic comparison is recommended to verify the function of BnaA9.NF-YA7. However, if ZS 11 is Hap3, the authors need to explain why the phenotypes are different between Fig. 1c and Fig. 2a. Is it due to inconsistent treatment conditions, e.g. different durations of drought stress? The treatment conditions should be clarified.

Response: In this study, the haplotype classification of ZS11 belongs to Hap3. We have added relevant content to the **Methods** section: **line 478**. ZS 11 is not heterozygous of Hap3 and Hap4, the omission of the name of Hap4 line has caused a misunderstanding, which we have addressed in the revised manuscript by including the name and relevant content as follows: **Methods** section, **lines 557-560**: “The promoter sequence of

BnaA9.NF-YA7 was amplified from ZS 11 (Hap3) and R219 (Hap4) and inserted into the binary vector pCAMBIA1305.1 to create *pBnaA9.NF-YA7^{Hap3}:GUS* and *pBnaA9.NF-YA7^{Hap4}:GUS* reporter gene constructs, respectively.”

As mentioned earlier, ZS 11 is Hap3. The visual differences between Fig. 1c and Fig. 2a are attributed to different treatment. In Fig. 1c, all lines were sourced from natural populations with unclear drought tolerance. Consequently, we implemented the longest possible drought treatment duration to discern their drought tolerance. Thus, the treatment condition for Fig. 1c involved subjecting the lines to 14-day drought treatment followed by 5-day re-watering period to observe survival rates. Because overexpression is highly susceptible to drought stress, we choose 7 days of drought treatment followed by 5 days of re-watering in Fig. 2a. This was done to visually highlight the difference in drought tolerance between WT and overexpression lines. We have provided additional details regarding these treatment conditions in the figure notes for panel Fig. 1c.

The revised place is as follows: **Figure legends** section, **lines 928-929**:

“c, The different varieties from five haplotypes survived after 14 days of drought treatment followed by 5 days of re-watering. Scale bars = 5 cm.”

2. The authors claimed that the co-existence of the M63I substitution and CCAAT-box deletion in *BnaA9.NF-YA7* is the cause of the strong drought tolerance of Hap4. However, this study mainly showed the effect of the CCAAT-box deletion on downregulating transcription of *BnaA9.NF-YA7*, while the influence of M63I was not convincingly demonstrated, because of the phenotypes of *BnaA9.NF-YA7G* and *BnaA9.NF-YA7C* OX lines in Fig. 2f are similar. Besides the difference in transcriptional activation of pBnA4.DOR detected in Fig. 6d, the authors should provide more experimental evidence to demonstrate the assumed important role of the M63I substitution.

Response: Thanks for your advice. We have added the self-activation test, the dual-luciferase transcriptional activity assays, and site-directed mutagenesis. We obtained fragments of different lengths of *BnaA9.NF-YA7*, the Y2H (self-activation) assay and the dual-luciferase reporter assay showed that *BnaA9.NF-YA7* was an activating transcription factor and the M63I substitution occurred in a Pro-rich transcriptional activation domain, resulting *BnaA9.NF-YA7* had different transcriptional activation abilities in Hap3 and Hap4. The results can be found in Supplementary Fig. 3d-h of the revised manuscript.

The corresponding content is as follows: **lines 98-104**

“Several *BnaA9.NF-YA7* fragments were designed based on the domain structure of *BnaA9.NF-YA7* (Supplementary Fig. 3d). The Y2H (self-activation) assay and dual-luciferase reporter assay showed that *BnaA9.NF-YA7* had different transcriptional activation abilities in Hap3 and Hap4 (Supplementary Fig. 3e-h), and the M63I substitution occurred in a Pro-rich transcriptional activation domain (Supplementary Fig. 3a).”

For site-directed mutagenesis, we introduced a site-directed mutation at the 189th base G of Hap3 by overlap extension PCR, resulting in substitution with C. Subsequently, we successfully obtained the M63I mutant expression vector under the control of Hap3 promoter. The transgenic *Arabidopsis* plants were generated through floral-dip by *Agrobacterium*-mediated and subjected to drought-rehydration experiment. The experimental findings are presented in Supplementary Fig. 8i-j: overexpression of both Hap3 and Hap4 heightened the sensitivity of transgenic plants towards drought stress. Notably, compared to Hap4 plants, those overexpressing Hap3 exhibited greater susceptibility to drought conditions; however, introduction of M63I site-directed mutagenesis restored their drought tolerance levels closer to that observed in Hap4 plants. In light of the results depicted in Fig. 2f as well, it is evident that although the M63I mutation can enhance drought tolerance relatively, overexpression *BnaA9.NF-YA7^C* still leads to increased sensitivity towards drought stress. This result is consistent with our speculated working model, *BnaA9.NF-YA7* can exert inhibitory effects on gene expression regulation by altering H3K36me3 levels, or act as a transcription activator alone. Taken together, *BnaA9.NF-YA7* plays a dual role as both an inhibitor and activator, effectively regulating ABA signaling in rapeseed to maintain a balance between growth promotion and drought stress responses.

The revised region is as follows: **Results** section: lines 145-146; **Methods** section: lines 526-531.

3. The authors stated that BnaA9.NF-YA7 has a feedback regulation on BnaABF3/4 by inhibiting the transcription of BnaABF3/4 through physically binding the BnaABF3/4 promoter as well as reducing the H3K36me3 levels. However, the following questions remain:

1) BnaA9.NF-YA7 directly binds the promoter of BnaABF3/4, does it directly inhibit the expression of ABF3/4? Transient dual-luciferase assays should be performed, like Fig. 4h and Fig. 6d.

Response: Thank you for your valuable advice. We have performed the transient dual-luciferase assays to investigate whether BnaA9.NF-YA7^G and BnaA9.NF-YA7^C directly inhibit ABF3/4 expression. As shown in Supplementary Fig. 18e (Supplementary Fig. 17 has been changed to Supplementary Fig. 18 in revised manuscript), although BnaA9.NF-YA7 directly binds to the promoter and CDS of ABF3/4, it does not directly inhibit the expression of ABF3/4. As a transcriptional activator (Supplementary Fig. 3d-h), overexpressing BnaA9.NF-YA7 induces weak LUC activity both the BnaC7.ABF3 and BnaC1.ABF4 reporters, and there is no difference between BnaA9.NF-YA7^G and BnaA9.NF-YA7^C.

2) RNA-seq and ChIP-seq showed that BnaA9.NF-YA7 directly binds the promoter of Bna.ASHH4 and represses its expression, and meanwhile, the H3K36me3 level of

BnaABF3 was decreased. However, whether the decreased expression of Bna.ASHH4 is the direct cause of the H3K36me3 level change of BnaABF3? More evidence should be provided for this conclusion.

Response: Thanks for your valuable advice. Due to the long construction period of complementation lines expressing *Bna.ASHH4s* sequences, we could not provide analysis of Bna.ASHH4s accumulation at *BnaC1.ABF3* and *BnaC7.ABF3* loci. Therefore, another strategy is used to support this conclusion. We constructed a VIGS vector of *Bna.ASHH4*, transformed into *Agrobacterium tumefaciens* (strain GV3101) and subsequently used for the infection of *B. napus* seedlings. In the lines where *Bna.ASHH4* expression levels were significantly down-regulated, we further detected the expression levels of *Bna.ABF3s* and *BnaA9.NF-YA7*. The results showed that the expression levels of *Bna.ABF3s* and *Bna.ASHH4* were significantly positively correlated. Even under the condition of dehydration, the expression level of *Bna.ABF3s* did not increase significantly in the virus-induced *Bna.ASHH4* silencing plants, and the expression level of *BnaA9.NF-YA7* was not significantly correlated with *Bna.ASHH4*. In addition, we also detected the H3K36me3 level of *Bna.ABF3* in TRV2:Bna.ASHH4 plants, and the results were consistent with those in BnaA9.NF-YA7-OE. These data further confirms that decreasing expression of *Bna.ASHH4* is the direct cause of the H3K36me3 level change of *Bna.ABF3s*.

The revised region is as follows: **Results** section: lines 331-344; **Methods** section: lines 521-523.

“Next, we performed virus-induced *Bna.ASHH4s* silencing using a tobacco rattle virus (TRV)-based system to reduce *Bna.ASHH4s* mRNA levels and determined the expression level of *BnaC1.ABF3* and *BnaC7.ABF3* in TRV2-*Bna.ASHH4* plants. The results indicated that *BnaC1.ABF3* and *BnaC7.ABF3* expression were positively correlated with *Bna.ASHH4s* expression. The expression of *BnaC1.ABF3* and *BnaC7.ABF3* in TRV2-*Bna.ASHH4* plants did not significantly increase even under dehydration conditions (Supplementary Fig. 21b). In turn, H3K36 trimethylation at *BnaC1.ABF3* and *BnaC7.ABF3* showed significant reduction in the TRV2-*Bna.ASHH4* lines (Supplementary Fig. 21c). In addition, the expression level of *BnaA9.NF-YA7* was not affected by the downregulation of *Bna.ASHH4s* under normal condition. Taken together, these results indicate that *BnaA9.NF-YA7* modulates the expression of *BnaABF3s* by controlling H3K36me3 levels.”

4. According to Fig. 4 and Supplementary Fig. 18, *BnaC7.ABF3* promotes the expression of *BnaA9.NF-YA7*, while the latter inhibits the expression of the former. When both of them were overexpressed in the DO transgenic line in Fig. 5f, how does the expression level of each gene change compared with WT? How to explain that *BnaA9.NF-YA7* acts downstream, but the drought tolerance of the DO line is significantly stronger than WT and more similar to *BnaC7.ABF3*-OX line?

Response: Thank you for your comment. We are sorry that we did not detect changes in the expression level of each gene in DO lines because we did not obtain stable genetic homozygous DO plants. In our experiment, we utilized homozygous *BnaC7.ABF3*-OE and *BnaA9.NF-YA7*-OE hybrids and subsequently employed F₁ generation for drought tolerance observation. However, due to multiple repeated drought tests, almost all F₁ hybrids were depleted, making it challenging for us to assess their expression levels in DO lines within a short timeframe. Based on the negative feedback regulation between *BnaA9.NF-YA7* and *BnaC7.ABF3*, as well as utilizing OE-7# (which exhibits the highest expression level in *BnaA9.NF-YA7* OEs) as the maternal parent, we speculate that overexpression of *BnaC7.ABF3* has limited effect on further increase of *BnaA9.NF-YA7* expression level. Referring to drought stress, the expression level of *BnaA9.NF-YA7* overexpressing plants will not further increase (Supplementary Fig. 19).

For “the drought tolerance of the DO line is significantly stronger than WT and more similar to *BnaC7.ABF3*-OX line”: Although *BnaA9.NF-YA7* is a direct downstream target of *Bna.ABF3s* and negatively feedback regulates their expression, *Bna.ABF3s* serve as a pivotal regulator in ABA signaling pathway, responsible for modulating the expression of numerous stress-responsive genes downstream. Meanwhile, while *BnaA9.NF-YA7* completely suppresses the expression of *BnaAn.ABF3* and *BnaC1.ABF3*, it only partially inhibits the expression of *BnaC7.ABF3*. We speculate that the partially inhibited expression level of *BnaC7.ABF3* in DO lines remains higher than that in WT, thereby contributing to the enhanced drought tolerance observed in DO lines than WT.

Other comments:

1. In the abstract, ‘two bases polymorphism’ should be ‘two base polymorphism’, and ‘a CCAAT cis elements’ should be ‘a CCAAT cis element’;

Response: Thank you very much for pointing out the mistake, ‘two bases polymorphism’ has been changed to ‘two base polymorphism’, and ‘a CCAAT cis elements’ has been changed to ‘a CCAAT cis element’ in the revision.

2. L225 ‘supplementary (Fig. 12c)’ should be ‘(Supplementary Fig. 12c)’;

Response: Supplementary Fig. 12 has been changed to Supplementary Fig. 13. ‘supplementary (Fig. 12c)’ has been changed to ‘(Supplementary Fig. 13c)’ in the revision.

3. Fig. 6j is mislabeled as Fig. 6f;

Response: This mistake has been corrected in the revision.

4. The discussion is largely a repetition of the results and should focus more on the findings' novel aspects.

Response: Thank you for your valuable advice. We have rewritten the discussion section, and detailed information can be found in the revised manuscript.

Reviewer #4 (Remarks to the Author):

In this paper, the authors identified that that the natural variation in BnaA9.NF-YA7 were associated with drought tolerance in *B. napa*. They revealed that BnaA9.NF-YA negatively regulated drought tolerance by suppressing the BnaABF genes and their downstream genes. In the drought-resistant traits, mutation of the CCAAT sequence in the BnaA9.NF-YA7 promoter causes low gene expression, and the M63I substitution decreased its transcriptional activity to the target genes. They concluded that these mutations could lead to less function of BnaA9.NF-YA7 and then decrease of the transpiration rate and water loss rate. The large amount of data supports that BnaA9.NF-YA7 negatively regulates desiccation tolerance in *B. rapa*. Negative transcriptional feedback with NF-Y factors in the ABA signaling pathway is very interesting. On the other hand, the contribution of the two mutations, especially the M63I mutation to drought resistance, is less clear, because most of the data were analyzed by using BnaA9.NF-YA7-overexpressing or -knockout plants. The authors need to show that this feedback regulation is different among the Hap. I'm not also sure how the M63I mutation affects expression of the BnaA9.NF-YA7-downstream gene in planta. The authors should analyze gene expression and drought tolerance in the BnaA9.NF-YA7 (Hap4) OX. In addition, there are some parts that the readers cannot follow due to lack of clear explanation (or low quality of the experiments) in the Results section and figures.

Response: Thank you for your comment. We have added the self-activation test (Supplementary Fig. 3d-f), the dual-luciferase transcriptional activity assays (Supplementary Fig. 3g, h), and site-directed mutagenesis (Supplementary Fig. 8i-j) to

demonstrate the assumed important role of the M63I substitution. In Fig. 8, we have displayed the difference in feedback regulation between haplotypes; however, it should be noted that the M63I mutation does not participate in feedback regulation. *BnaA9.NF-YA7* attenuated stoma sensitivity to ABA by activating downstream gene *Bna.DORs*, thereby ensuring the influx of CO₂ for photosynthesis. The presence of the M63I mutation within the TA region leads to a reduction in *Bna.DORs* expression and stomata closure. Nevertheless, the effect of the M63I mutation on drought tolerance is limited and only when accompanied by the CCAAT mutation can further enhance drought tolerance. This may also explain why Hap4 exhibits stronger drought tolerance than Hap1. Therefore, we deduce that these two mutations are both independent yet tightly linked; with CCAAT primarily influencing feedback regulation loop while M63I predominantly modulates stoma sensitivity towards ABA. In Fig. 2f, we conducted an analysis on the drought tolerance of *BnaA9.NF-YA7^C-OE* (Hap4) plants and found that the survival rate after drought treatment did not show a significant difference compared to *BnaA9.NF-YA7^G-OE* lines. Additionally, we examined the expression levels of *BnaC1.ABF3*, *BnaC7.ABF3*, *BnaC1.ABF4*, *BnaC6.ASHH4*, and *BnaA4.DOR* in *BnaA9.NF-YA7^C-OE* plants. Although the difference in *BnaA9.NF-YA7* expression levels between *BnaA9.NF-YA7^G-OE* and *BnaA9.NF-YA7^C-OE* lines impeded the direct comparison, the findings revealed no significant divergence in the expression patterns of other genes, except for a notable reduction in *BnaA4.DOR* expression level. (Supplementary Fig. 19).

In addition, the content of the **Results** section has been further enhanced and supplemented with explanations for figures, aiming to facilitate readers' comprehension of the research findings.

In Line 66, the authors described “Further mining previously published data showed that ...”, but detail of the mining was not explained. Why was *BnaA9.NF-YA7* newly found as a good candidate?

Response: In the previously published data, we used 60k chip data. Subsequently, resequencing data was obtained, and a genome-wide association analysis was conducted using previously published phenotype data combined with resequencing data. Then some new candidate genes were identified. Based on the transcriptome data of repeated drought-rewatering treatment obtained in the laboratory in the early stage, we found that *BnaA9.NF-YA7* strongly responds to drought stress (Supplementary Fig. 12a), so we will use it as a key candidate gene for research.

There are some cases where the plant appearance in the photo does not match the survival rate in the drought tolerance tests. For example, in Fig. 1c, some leaves are healthy on Hap2, whereas all leaves in Hap3 and Hap5 plants seemed to be shriveled. However, in Fig1d, Hap2 and Hap5 have the same survival rate, and Hap3 has a lower survival rate. Why do these differences occur? Please clarify the criteria for determining whether the plants are survived or dead.

Response: For survival rate of the five Haps, we calculated the mean of all accessions in each haplotype. Based on the average value and survival rate distribution of each haplotype, it was judged whether the haplotype was drought-tolerant. However, each haplotype contains multiple accessions. Drought, as a typical quantitative trait, is regulated by multiple genes, even drought sensitive haplotypes still have some drought tolerant accessions. We didn't select the most extremely different phenotype subjectively when we are taking photos.

In Supplementary Fig 9a, I'm not sure where the fluorescence of YFP and DAPI are localized. Please show more clear images.

Response: Thank you for your valuable advice, we have provided clearer images in the revised manuscript. The fluorescence of YFP endodermis was enriched in endodermis and vascular tissue of root tip maturation zone. The fluorescence localization results have limited reference value due to the inadequate staining effect of DAPI in supplementary experiments.

There are other NF-Ys than BnaA9-B2 and BnaA6-C9 in the B2 subgroup of Supplementary Fig. 11a. Why did the authors test the combination of BnaA9-B2 and two C9s in Y2H, and those of two NF-YAs in Y3H? I'm also not sure how to identify NF-YB2 and NF-YCs that could form a trimer complex with BnaA9.NF-YA7 in Supplementary Fig. 16.

Response: We tested all Bna.NF-YB/YC combinations in the B2 subgroup (six Bna.NF-YBs and five Bna.NF-YCs), but only BnaA9.NF-YB2/BnaA6.NF-YC9 and BnaA9.NF-YB2/BnaC5.NF-YC9 combinations had interactions. Then, we conducted Y3H analysis using two combinations (BnaA9.NF-YB2/BnaA6.NF-YC9 and BnaA9.NF-YB2/BnaC5.NF-YC9) and 12 Bna.NF-YAs in the B2 subgroup, except for BnaA9.NF-YA7 and BnaC5.NF-YA7. Among these Bna.NF-YAs, only BnaA6.NF-YA5 and BnaC5.NF-YA9 had significant interaction with two YB/YC combinations. In the Supplementary Fig. 12a (Supplementary Fig. 11 has been changed to Supplementary Fig. 12), we only showed YB/YC combinations and NF-Y combinations with interaction relationships. For Supplementary Fig. 17 (Supplementary Fig. 16 has been changed to Supplementary Fig. 17), we first examined the pairwise interaction between the three subunits of NF-Y and found that there is a strong interaction between Bna.NF-YB and Bna.NF-YC, but BnaA9.NF-YA7 did not interact with Bna.NF-YB directly or existed a weak interaction (Supplementary Fig. 17a-b). Then, the Bna.NF-YB protein is cloned in MCSI and expressed as a fusion to the GAL4 DNA-BD, while the Bna.NF-YC protein is cloned in MCSII and conditionally expressed from the *P_{Met25}* promoter. If the conditional expression of Bna.NF-YC protein allows the clone to survive on SD/-H-L-M-W medium and is stronger than Y2H, it indicates that Bna.NF-YBs and Bna.NF-YCs that could form a trimer complex with BnaA9.NF-YA7 (Supplementary Fig. 17c). Positive interactions that activate the Aureobasidin A (AbA) resistance reporter gene are detected by plating on selective media containing AbA. To further determine the role of Bna.NF-YCs in the formation of trimers, positive clones were plated onto SD/-Leu/-Met/-Trp/AbA and SD/-Leu/-Trp/AbA media, respectively. Colonies that do not survive on the SD/-Leu/-

Trp/AbA plating contain candidate prey proteins that may be part of a productive ternary interaction involving the bridge protein and the BD bait (Supplementary Fig. 17d).

I couldn't follow what Supplementary Fig. 16d meant. Why did addition of Met cause inhibition of yeast growth?

Response: Supplementary Fig. 16 has been changed to Supplementary Fig. 17 in revised manuscript. For the yeast three-hybrid assay, the pGADT7 vector was used to express BnaA9.NF-YA7 and the pBridge vector was used to express both BnaNF-YBs and BnaNF-YCs. Wherein, the BnaNF-YB coding sequence was cloned into the EcoRI and BamHI sites of multiple cloning site I (MCSI) and the BnaNF-YC (BnaC8.NF-YC1 or BnaA6.NF-YC4) coding sequence was cloned into the NotI and BglII sites of MCSII of the pBridge vector. **Expression of the third protein (BnaC8.NF-YC1 or BnaA6.NF-YC4) is controlled by a conditional methionine repressible promoter (P_{Met25}), i.e. it is expressed when methionine is absent from the growth medium.** In general, three-hybrid need to determine if the 3rd protein acts as a bridge or enhancer of bait-prey interactions. First, perform a two-hybrid screen using pBridge, but plate on SD/-Leu/-Met/-Trp/AbA media to express both bait proteins while selecting for AbA-resistance. Next, patch-out the positive clones on selective, methionine-containing media that represses expression of the bridge protein (e.g. SD/-Leu/-Trp/AbA). If colonies that do not survive the second plating contain candidate prey proteins that may be part of a productive ternary interaction involving the bridge protein and the BD bait.

Minor points,

Why do yeast colonies in the left panel of Supplementary Fig.16d look blue?

Response: We deeply sorry for the confusion, this visual difference is caused by the varying brightness of the shooting light. Carefully observing the culture medium and clones of Supplementary Fig.17d (Supplementary Fig. 16 has been changed to Supplementary Fig. 17 in revised manuscript) and Supplementary Fig.17c, you will find that the blue color presented by Supplementary Fig.17d is significantly different from the blue color produced by protein interactions in Supplementary Fig.17c.

The one-letter abbreviation of tryptophan is W.

Response: Thank you very much for pointing out the mistake, they've been corrected in the revision.

Measurement of the soluble protein and sugar content, and other physiological indicators that were written in the Methods, were not performed in the manuscript.

Response: We are sorry for this mistake. The soluble protein, proline content, RWC, and seedling fresh weight can be found in Supplementary Fig.2e-h. We carefully checked the manuscript, some physiological indicators (such as soluble sugar, MAD, SOD, etc.) not shown in the results were removed. We have revised the content of the Methods. The revised region is as follows: **Methods** section, **lines 537-548**.

Method of the FRET analysis was not found in the manuscript.

Response: A brief description of the method for BiFC-FRET analysis added to the revision:

“For BiFC-based FRET analysis, the coding regions of *BnaA9.NF-YB2* and *BnaA6.NF-YC4* were cloned and inserted into the pXY106 vector and pXY104 vector to produce nYFP-BnaA9.NF-YB2 and cYFP-BnaA6.NF-YC4, respectively. The full-length sequences of *BnaA9.NF-YA7^G* and *BnaA9.NF-YA7^C* were amplified, and the N-terminus was fused to the CFP coding sequence, generating BnaA9.NF-YA7^G-CFP and BnaA9.NF-YA7^C-CFP, respectively. The plasmids were then transformed into *Agrobacterium* GV3101 cells with the silencing suppressor P19 strain. Then, *Agrobacterium* cultures containing equal amounts of nYFP-BnaA9.NF-YB2 and cYFP-BnaA6.NF-YC4 strains were mixed and infiltrated into *Nicotiana benthamiana* leaves to allow the formation of BiFC complexes (acceptors), along with equal amounts of BnaA9.NF-YA7^G-CFP or BnaA9.NF-YA7^C-CFP (donors) strain cultures. After infiltration, the plants were grown at 22°C under a 12-h light/12-h dark cycle in a greenhouse for 48 to 72 h before analyses of interaction using FRET. An excitation wavelength of 458 nm and an emission wavelength of 470–500 nm was used for CFP, and an excitation wavelength of 514 nm and an emission wavelength of 515–545 nm were used for YFP. The FRET energy transfer efficiency was calculated using the formula $E = (\text{donor after} - \text{donor before}) \times 100 / \text{donor after}$, and is shown as a percentage^{51,52}.”

Reviewers' Comments:

Reviewer #2:

Remarks to the Author:

Thank you for the substantial revisions based on my comments and other reviewers.

In revision of water holding capacity (WHC), the term relative water content (RWC) is introduced in Methods. RWC is not used anywhere else in the main manuscript. Is it used elsewhere in Supplementary information? Is this an error?

Reviewer #3:

Remarks to the Author:

The authors have satisfactorily addressed my comments.

Reviewer #4:

Remarks to the Author:

In the revised manuscript, the authors addressed some issues that the reviewer1 and I raised, but unfortunately there are still some big concerns, because they didn't respond to the important comment, or explain their revision in the manuscript and Responses. In addition, although the authors wrote "BnaA9.NF-YA7 related NF-Y complex physically associate with the BnaABF3/4 promoter and coding region to downregulate their expression by controlling H3K36me3 levels" in the Abstract and depicted the working model in Fig. 8, new Supplemental Fig. 18e and Supplemental Fig. 21a weakened the relationship between the binding of BnaA9NF-YA7 to the genomic region and their H3K36me3 levels for the suppression of their gene expression. I think that the suppression of Bna.ABF3/4 expression should only be explained by indirect regulation by H3K36me3 via Bna.ASHH4s. There remains a possibility that just artificial binding was detected due to the ChIP-seq using the NF-YA7 overexpressors and EMSA analyses.

Moreover, I understand that the figures and tables in the manuscript consist of a part of numerous experiments that the authors conducted, and it would be difficult to present all of the data. However, the data and explanations in the revised paper are also too limited for readers to properly understand them. It is not adequate responses that the authors only provided detailed information (or excuses) in the Response to the reviewers, but do not revise descriptions or add new figures and tables (even in a form of supplemental data).

I found no descriptions and answers about the comparison of H3K36me3 with Hap3/4, pointed out by Reviewer 1.

The authors wrote that the survival rate in Fig. 1c was quite different from that in Fig. 1d, because Fig. 1d is the average of different genotypes in each haplotype. I think that the survival rate of all genotypes should be shown similar to Supplementary Fig. 2e-h (The results of one experiment out of five times would be OK). In addition, I agree with the authors' opinion that the most extremely different individuals/genotypes should not be subjectively chosen. However, I think that readers will generally consider the photographic data to be representative of the experiment. Why did the authors show these individuals/genotypes as the representative ones? I ask the same comment and question about Supplemental Fig.5c.

Are transgenic plants in new Supplemental Fig.8i, j Arabidopsis? There is no description in the text or figure legend.

In Supplemental Fig. 10a, new photos of protoplasts have been added but there is no explanation

what they are in the text, figure legend, or Response. "Marker" signals are located at the same location as GFP in both photos. What do they indicate? Furthermore, it is still impossible to see whether the YFP and DAPI signals overlap in root vasculatures. I think it would be better to take more zoomed photos or delete them.

Blue colonies in the X-gal assay look very different among Supplemental Fig.12c, Fig.17c, and Fig.18a. The excuse that the colors are different between Supplemental Fig.17c and d is not acceptable.

REVIEWER COMMENTS

Reviewer #2 (Remarks to the Author):

Thank you for the substantial revisions based on my comments and other reviewers.

In revision of water holding capacity (WHC), the term relative water content (RWC) is introduced in Methods. RWC is not used anywhere else in the main manuscript. Is it used elsewhere in Supplementary information? Is this is an error?

Response: Sincerely appreciate your constructive comments again. The manuscript primarily utilized WHC as the phenotype, as it better reflects the drought tolerance differences among transgenic plants compared to RWC. RWC was only employed in the analysis of physiological indicators for the five haplotypes and you can find RWC in **Supplementary Fig. 2f**.

Reviewer #3 (Remarks to the Author):

The authors have satisfactorily addressed my comments.

Response: Sincerely appreciate your constructive comments again.

Reviewer #4 (Remarks to the Author):

In the revised manuscript, the authors addressed some issues that the reviewer1 and I raised, but unfortunately there are still some big concerns, because they didn't respond to the important comment, or explain their revision in the manuscript and Responses. In addition, although the authors wrote "BnaA9.NF-YA7 related NF-Y complex physically associate with the BnaABF3/4 promoter and coding region to downregulate their expression by controlling H3K36me3 levels" in the Abstract and depicted the working model in Fig. 8, new Supplemental Fig. 18e and Supplemental Fig. 21a weakened the relationship between the binding of BnaA9NF-YA7 to the genomic region and their H3K36me3 levels for the suppression of their gene expression. I think that the suppression of Bna.ABF3/4 expression should only be explained by indirect regulation by H3K36me3 via Bna.ASHH4s. There remains a possibility that just artificial binding was detected due to the ChIP-seq using the NF-YA7 overexpressors and EMSA analyses.

Moreover, I understand that the figures and tables in the manuscript consist of a part of numerous experiments that the authors conducted, and it would be difficult to present all of the data.

However, the data and explanations in the revised paper are also too limited for readers to properly understand them. It is not adequate responses that the authors only provided detailed information (or excuses) in the Response to the reviewers, but do not revise descriptions or add new figures and tables (even in a form of supplemental data).

"In the revised manuscript, the authors addressed some issues that the reviewer1 and I raised, but unfortunately there are still some big concerns, because they didn't respond to the important comment, or explain their revision in the manuscript and Responses."

Response: We are sorry for the inconvenience in the revised manuscript. We sincerely thank reviewers for the constructive suggestions, and time for reviewing our manuscript. We will try our best to improve these omitted problems in the revised manuscript.

“In addition, although the authors wrote “BnaA9.NF-YA7 related NF-Y complex physically associate with the BnaABF3/4 promoter and coding region to downregulate their expression by controlling H3K36me3 levels” in the Abstract and depicted the working model in Fig. 8, new Supplemental Fig. 18e and Supplemental Fig. 21a weakened the relationship between the binding of BnaA9NF-YA7 to the genomic region and their H3K36me3 levels for the suppression of their gene expression. I think that the suppression of Bna.ABF3/4 expression should only be explained by indirect regulation by H3K36me3 via Bna.ASHH4s. There remains a possibility that just artificial binding was detected due to the ChIP-seq using the NF-YA7 overexpressors and EMSA analyses.”

Response: Thank you very much for the comments, and we totally agree “BnaA9NF-YA7 suppresses of Bna.ABF3/4 expression by indirect regulation by H3K36me3 via Bna.ASHH4s”, and we have corrected this in the whole manuscript.

“Moreover, I understand that the figures and tables in the manuscript consist of a part of numerous experiments that the authors conducted, and it would be difficult to present all of the data. However, the data and explanations in the revised paper are also too limited for readers to properly understand them. It is not adequate responses that the authors only provided detailed information (or excuses) in the Response to the reviewers, but do not revise descriptions or add new figures and tables (even in a form of supplemental data).”

Response: Sincerely thank you for your suggestion, we carefully checked the manuscript and include all the response results to the main text or supplementary files. For “how to identify NF-YB2 and NF-YCs that could form a trimer complex with BnaA9.NF-YA7” and “Why did addition of Met cause inhibition of yeast growth” involve the experimental principle of yeast three-hybridization, and the method section introduces the specific experimental steps, so we only explain to you. For “Why did the authors test the combination of BnaA9-B2 and two C9s in Y2H, and those of two NF-YAs in Y3H”, the yeast two-hybrid results involving six Bna.NF-YBs and five Bna.NF-YCs from B2 subgroup have been added to Supplementary Fig. 12b, and the yeast three-hybrid results encompassing the dimers BnaA9.NF-YB2/BnaA6.NF-YC9 and BnaA9.NF-YB2/BnaC5.NF-YC9 along with 12 Bna.NF-YAs from B2 subgroup have been added to Supplementary Fig. 12d.

b

d

I found no descriptions and answers about the comparison of H3K36me3 with Hap3/4, pointed out by Reviewer 1.

Response: Thank you for pointing out the omission in our work. We have supplemented results of the comparison of H3K36me3 with Hap3/4. The revised regions are as follows: **Results** section, **lines 340-344**.

‘Moreover, ChIP analysis of H3K36 trimethylation in Hap3 and Hap4 genotypes revealed that high expression level of *BnaA9.NF-YA7* significantly reduced the extent of H3K36 trimethylation at *BnaC1.ABF3* and *BnaC7.ABF3* loci but not *BnaA4.DOR* and *BnaA2.ACT7* loci (Supplementary Fig. 22).’

The authors wrote that the survival rate in Fig. 1c was quite different from that in Fig. 1d, because Fig. 1d is the average of different genotypes in each haplotype. I think that the survival rate of all genotypes should be shown similar to Supplementary Fig. 2e-h (The results of one experiment out of five times would be OK). In addition, I agree with the authors' opinion that the most extremely different individuals/genotypes should not be subjectively chosen. However, I think that readers will generally consider the photographic data to be representative of the experiment. Why did the authors show these individuals/genotypes as the representative ones? I ask the same comment and question about Supplemental Fig.5c.

Response: Thank you for your comment. To avoid reader's confusion, we selected a photograph close to the statistical results from the original photos to replace the Fig 1c picture, and we also performed the same operation on Supplemental Fig.5c. The results after replacement are as follows:

Are transgenic plants in new Supplemental Fig.8i, j *Arabidopsis*? There is no description in the text or figure legend.

Response: Thank you for your comment. The transgenic plants in Supplemental Fig.8i, j were *Arabidopsis*, you can find the relevant description in the **Methods** section: **lines 529-534**. In addition, we have added relevant descriptions in the figure legend of the Supplementary information.

In Supplemental Fig. 10a, new photos of protoplasts have been added but there is no explanation what they are in the text, figure legend, or Response. "Marker" signals are located at the same location as GFP in both photos. What do they indicate? Furthermore, it is still impossible to see whether the YFP and DAPI signals overlap in root vasculatures. I think it would be better to take more zoomed photos or delete them.

Response: Thank you for your comment. The new photos of protoplasts serve as a supplementary illustration to address your uncertainty regarding the subcellular localization results of YFP and DAPI, which have been thoroughly explained in the original manuscript (**lines 170-172**, the relevant content has been deleted from the revised manuscript). "Marker" indicates mCherry. Considering that the subcellular localization results have no substantial contribution to the experimental results, we have decided to accept your suggestion and remove the relevant content from the manuscript. The revised Supplemental Fig. 10 only retains the GUS staining results.

Blue colonies in the X-gal assay look very different among Supplemental Fig.12c, Fig.17c, and Fig.18a. The excuse that the colors are different between Supplemental Fig.17c and d is not acceptable.

Response: When conducting the yeast experiments in Supplemental Fig. 12c and 17c, we inadvertently omitted the addition of X- α -gal to the medium. To address this oversight, we implemented a strategy of coating X- α -gal on the surface of the medium for these experiments. As

a result, there is a slight disparity in the coloration of the blue clones observed in Supplemental Fig. 12c and 17c compared to those depicted in Supplemental Fig. 18a (where X- α -gal was added directly to the medium). For Supplemental Fig. 17d, we redo the yeast experiment, and the new results were used to replace the original image. The details can be found in revised Supplementary Figure 17d.

Reviewers' Comments:

Reviewer #4:

Remarks to the Author:

In the revised manuscript, the authors adequately addressed all the issues I raised. Finally, I think that the authors should revise Fig. 8, because it still shows as if NF-Y complex rather than BnaA SHH4s would directly regulate H3K36me3 levels.

REVIEWER COMMENTS

REVIEWERS' COMMENTS

Reviewer #4 (Remarks to the Author):

In the revised manuscript, the authors adequately addressed all the issues I raised. Finally, I think that the authors should revise Fig. 8, because it still shows as if NF-Y complex rather than BnaA SHH4s would directly regulate H3K36me3 levels.

Response: Thank you very much for the comments, and we have revised Fig. 8 to address your suggestion.